# R2E-Gym: Procedural Environments and Hybrid Verifiers for Scaling Open-Weights SWE Agents

**Naman Jain**[1][*] **Jaskirat Singh**[2][*] **Manish Shetty**[1] **Tianjun Zhang**[1]
**Liang Zheng**[2] **Koushik Sen**[1] **Ion Stoica**[1]

[1]UC Berkeley [2]Australian National University

{naman_jain@berkeley.edu jaskirat.singh@anu.edu.au}

## Abstract

Improving open-source models on real-world SWE tasks (solving GITHUB issues) faces two key challenges: 1) scalable curation of execution environments to train these models, and 2) optimal scaling of test-time compute. We introduce R2E-Gym, the largest procedurally-curated executable gym environment for training real-world SWE-agents, consisting of more than 8.1K tasks. R2E-Gym is powered by two main contributions: 1) SWEGEN: a synthetic data curation recipe that enables scalable curation of executable environments using test-generation and back-translation directly from commits, thereby reducing reliance on human-written issues or unit tests. We show that this enables more scalable training leading to PASS@1 of 34.4% on SWEBENCH-VERIFIED benchmark with our 32B model. 2) Hybrid Test-time Scaling: we next provide an in-depth analysis of two test-time scaling axes; execution-based and execution-free verifiers, demonstrating that they exhibit complementary strengths and limitations. Test-based verifiers suffer from low distinguishability, while execution-free verifiers are biased and often rely on stylistic features. Surprisingly, we find that while each approach individually saturates around 42-43%, significantly higher gains can be obtained by leveraging their complementary strengths. Overall, our approach achieves **51**% on the SWEBENCH-VERIFIED benchmark, reflecting a new state-of-the-art for open-weight SWE agents and for first time being competitive with proprietary systems such as o1 or sonnet w/ tools.

## 1 Introduction

Autonomous software engineering (SWE), aiming to solve real-world software engineering problems such as GITHUB issues, has made significant progress in recent times (Wang et al., 2024; Yang et al., 2024b). While LLM-based SWE-Agents have demonstrated remarkable improvements, state-of-the-art performance is largely driven by proprietary models (Anthropic, 2025; Jaech et al., 2024) — with open-models lagging behind (Xie et al., 2025).

Addressing this gap requires solving two fundamental challenges: First, scalable curation of high-quality execution environments to train these models; and second, developing efficient aggregation strategies to maximize test-time performance. While several benchmarks for evaluating SWE-agents on GITHUB issues exist (Jimenez et al., 2023; Zhao et al., 2024), scalable curation of high-quality training environments remains a challenging problem. For instance, while the training split from SWE-Bench (Jimenez et al., 2023) contains output patches, it lacks executable environments. Pan et al. (2024) collect executable test environments, but rely on human-written issues and test cases restricting sample-size.

In this paper, we introduce R2E-GYM, the largest procedurally curated environment for training real-world SWE-agents — consisting of more than 8.1K problems, with executable

---

[*] Equal Contribution.

**Project Page:** https://r2e-gym.github.io

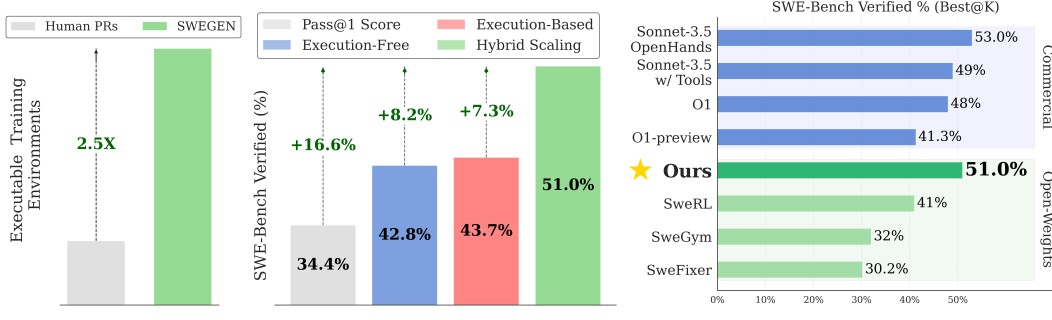

Figure 1: **Overview.** In this paper, we introduce R2E-Gym, the largest gym environment and training framework for training open-weight SWE agents. R2E-Gym is powered by two main contributions: (a) SWEGEN: a synthetic data curation recipe for curating executable training environments w/o relying on human tests and issues (§2). (b) Hybrid Inference Time Scaling: showing that while both execution-based and execution-free verifiers elicit inference-time gains; significantly better performance can be achieved by leveraging the strengths of both (§4). (c) Overall, the final approach reflects SOTA performance for open-weight SWE-agents, while also being competitive with some proprietary model baselines[1].

gym environments, unit tests, and natural-language task descriptions (§2). R2E-GYM addresses both key challenges through two primary contributions (Figures 1a and 1b):

**Synthetic Data Enables More Scalable Training.** We propose SWEGEN — a novel synthetic data curation recipe that enables collection of a large number of executable training environments without reliance on human-written pull requests (PRs) or unit tests. We show that instead of using human-written PRs, good-quality execution environments can directly be curated from *commits* through backtranslation (Li et al., 2023; Wei et al., 2023) and test collection or generation (§2). Compared to PR-based data collection (Pan et al., 2024), this approach enables more scalable data curation (Figure 1a) and agent-training, resulting in a PASS@1 performance of 34.4% on the challenging SWEBENCH-VERIFIED benchmark.

**Hybrid Inference Time Scaling.** We next leverage R2E-GYM to investigate two complementary axes for scaling test-time compute (§4): 1) Execution-based verifiers that evaluate patches through test cases (Xia et al., 2024b), and 2) Execution-free verifiers that assess trajectories through learned models (Pan et al., 2024). While prior works have studied these approaches in isolation, they lack a comprehensive analysis of their relative strengths and weaknesses. We first present a unique and in-depth analysis of their working mechanisms, demonstrating that execution-free and execution-based methods actually exhibit complementary strengths and weaknesses. We find two key insights (studied in §4.2): a) Execution-based methods provide direct signals for patch correctness but struggle with discriminating between solutions, and b) Execution-free verifiers provide better discrimination but can be biased by other heuristics (*e.g.*, agent thoughts) over the final patch. Based on the above insights, we propose a hybrid scaling approach leveraging the strengths of both methods. Surprisingly, while the performance of both execution-based and execution-free methods plateaus around 42-43%, the hybrid approach yields significantly higher gains, achieving a final performance of 51% on SWEBENCH-VERIFIED (Figure 1b and §4.3).

The key contributions of this paper are: 1) We introduce R2E-GYM, the largest procedurally curated environment for training real-world SWE-agents, increasing the number of executable environments by over 3 times. 2) We provide an in-depth analysis demonstrating that execution-based and execution-free axes for scaling test-time compute exhibit complementary strengths and weaknesses. 3) Based on the above insights, we propose a *hybrid scaling* approach that leverages the strengths of both methods, significantly improving test-time performance. 4) Finally, we release an open-weights 32B model that achieves 51% on SWEBENCH-VERIFIED, reflecting a new state-of-the-art for open-weight SWE-agents, while also for the first time demonstrating competitive or better performance compared to commercial models (Fig. 1c), e.g., o1 (Jaech et al., 2024) and sonnet-3.5-v2 (Anthropic, 2024).

---

[1]Results with all open-weight models are reported with test-time scaling.

## 2 R2E-GYM: Procedural Synthetic Data Generation

| Dataset (split) | Repo? | Executable? | # Instances |
|---|---|---|---|
| APPS (Hendrycks et al., 2021) | ✗ | ✓ | 10,000 |
| R2E (Jain et al., 2024b) | ✓ | ✓ | 246 |
| SWE-Bench(train) (Jimenez et al., 2023) | ✓ | ✗ | 19,008 |
| SWE-Gym Raw (Pan et al., 2024) | ✓ | ✗ | 66,894 |
| SWE-Bench (test) (Jimenez et al., 2023) | ✓ | ✓ | 2,294 |
| SWE-Gym (Pan et al., 2024) | ✓ | ✓ | 2,438 |
| R2E-Gym-Subset (**Ours**) | ✓ | ✓ | 4,578 |
| R2E-Gym (**Ours**) | ✓ | ✓ | **8,135** |

Table 1: **Dataset Statistics.** Comparing statistics across different datasets curating executable training environments for SWE-agent training. R2E-Gym refers to our full dataset, and R2E-Gym-Subset refers to a filtered subset of tasks, with non-overlapping repositories with SWE-Bench.

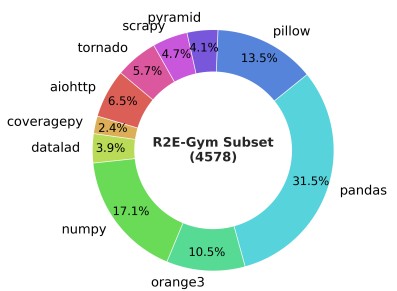

Table 2: **Repo distribution** for R2E-Gym subset (no overlap with SWE-Bench) used for training (refer §3).

**Overview.** SWE task collection methods (Jimenez et al., 2023) rely on human-written issues and unit tests for problem statements and evaluation functions. However, this presents a challenge for scaling data curation as size is limited by human-written PRs. To overcome this limitation, we propose SWEGEN — a synthetic data curation recipe using backtranslation and test generation. We procedurally generate environments using only commits from GITHUB repositories, reducing reliance on both human-written issues and test cases.

**Repository and Commit Curation.** We use SEART GITHUB search[2] to identify PYTHON repositories with a large number of commits. Next, we extract commit history and associated code changes for each repository. We filter relevant commits using a combination of rule-based and LLM-based heuristics, identifying *interesting* code changes. For each relevant commit, we next collect build scripts by semi-manually searching across dependency pins. We expand our set of heuristics and installation procedure further in the Appendix A.

**Test-Validation and Generation for Environment Collection.** Following Jimenez et al. (2023), we use the existing test cases in the curated commits to identify Fail→Pass (F2P) test cases, i.e. test cases that fail in the original buggy commit and pass in the fixed commit. In cases where the curated commits do not have associated tests, limiting the ability to use them for training environments, we supplement such commits with automatically generated Fail→Pass test-cases. Appendix A expands our test generation approach.

**Backtranslation: Non-reliance on GITHUB Issues.** Using the above steps, we collect a large number of commits, associated build environments and F2P (Fail→Pass) test cases. Now, we need to collect the problem statements associated with the commits. Prior works (Jimenez et al., 2023; Pan et al., 2024) use human-written GITHUB issues as problem statements. This inevitably cannot use the entire commit history since human-written issues are not available for all commits. Here, following Li et al. (2023); Wei et al. (2023) we propose a backtranslation approach to collect the problem statements associated with the commits.

However, naively back-translating code changes is quite noisy as models often generate generic problem statements that do not capture the essence of the code changes. Instead, we identify that human-written issues often contain failing tests and execution traces as part of bug reports. We use this observation to collect high-quality problem statements by using the F2P test-cases as part of the backtranslation prompt. Similar to existing works (Jain et al., 2024b; Zhuo et al., 2024), we find that using test execution information allows generating precise and directed problem statements. Please find prompts and examples in Appendix.

We collect over 8.1K problem statements using this approach (referred to as R2E-Gym). We decontaminate this set by removing repositories overlapping with SWE-Bench test-set repositories, obtaining 4578 problems (referred to as R2E-Gym-Subset) and use that across all experiments unless specified otherwise. Table 1 shows the statistics of different datasets, and Figure 2 and Figure 9 show the distribution of the repositories in R2E-Gym-Subset and

---

[2]https://seart-ghs.si.usi.ch/

Table 3: Resolve Rate (%) Comparison on SWEBENCH-VERIFIED and SWEBENCH-LITE benchmarks. We observe that synthetic data curation (SWEGEN): allows our approach to scale better across different model sizes. All experiments use the `Qwen-2.5-Coder` as base-models.

| Model Size | SWEBENCH-LITE | | | | SWEBENCH-VERIFIED | | | |
|---|---|---|---|---|---|---|---|---|
| | **Base-model** | **SWE-Gym** | **Ours** | Δ | **Base-model** | **SWE-Gym** | **Ours** | Δ |
| 7B | 1.0 (±1.0) | 10.0 (±2.4) | **11.0** (±0.8) | +1.0 | 1.8 (±1.3) | 10.6 (±2.1) | **19.0** (±1.0) | +8.4 |
| 14B | 2.7 (±1.9) | 12.7 (±2.3) | **20.67** (±0.7) | +7.97 | 4.0 (±1.6) | 16.4 (±2.0) | **26.8** (±1.4) | +10.4 |
| 32B | 3.0 (±1.4) | 15.3 (±2.5) | **23.77** (±0.8) | +8.47 | 7.0 (±1.3) | 20.6 (±2.1) | **34.4** (±1.2) | +13.8 |

R2E-Gym respectively. Notably, using our SWEGEN approach, we can collect over 2.5 times more problems than relying on the data collection relying on GITHUB issues (Figure 1a).

# 3 Training SWE-Agents using R2E-GYM Environments

**Agent Scaffolding.** We design a minimal scaffold on top of OPENHANDS (Wang et al., 2024) to experiment with agents for diverse SWE tasks. It uses a traditional REACT framework (Yao et al., 2022) without any specialized workflow; equipping the LLM with only a bash terminal, file editor, and search tool. Figure 16 depicts an example code editing trajectory.

**Trajectory Collection and SFT Training**. We next collect SFT trajectories using from R2E-Gym environments. To avoid contamination, we only use a subset of R2E-Gym consisting of repos with no overlap with the SWE-Bench dataset. The resulting subset (R2E-Gym-Subset) consists of 4578 executable environments across 10 repositories (Figure 2). For each task environment, we use SONNET-3.5-V2 with our agent scaffold and collect the successful agent trajectories. Through this process, we collect 3321 trajectories from 2048 unique task environments. We then use these trajectories to train our agent via supervised fine-tuning on agent thoughts and actions. For training, we use LLaMA-Factory (Zheng et al., 2024) and `Qwen-2.5-Coder` models (7B, 14B, 32B) as our base models. For detailed experiment configuration and hyperparameters, please refer to Appendix B.

## 3.1 Results and Analysis

**Comparison to open-weight SWE-Agents across Model Scales**. We report PASS@1 of R2E-Gym trained models on the SWEBENCH-VERIFIED and SWEBENCH-LITE benchmarks in Table 3. We also report comparisons with recently proposed SWE-Gym (Pan et al., 2024), which is most closest to our work. As seen in Table 3, we find that our approach enables better scaling for training SWE-agents across all model sizes. For instance, on SWEBENCH-VERIFIED, for the same base-model type and scale, our 32B model significantly improves the PASS@1 performance by 14%; pushing the final performance from 20.6 (SWE-Gym) to 34.4%.

**Scaling with Number of Trajectories**. We investigate the relationship between training samplesize (number of trajectories) and agent performance in Figure 2. We evaluate 14B and 32B models trained with trajectory counts ranging from 100 to 3,200. Our findings indicate that performance improves with increasing trajectory count, though with diminishing returns for both models. Notably, the 14B model begins to saturate at approximately 800 samples, while the 32B model still shows improvements, likely due to its larger capacity. These results extend the findings of Pan et al. (2024), who studied dataset scaling up to ∼ 500 samples. Our analysis demonstrates that while performance does improve with increasing samplesize, the rate of improvement diminishes or even plateaus for smaller models.

**Real vs Synthetic Problem Statements.** The R2E-Gym approach enables us to generate problem statements without relying on human-written descriptions and test cases, offering greater scalability. We compare the performance of models trained on real GitHub issues versus our synthetic problem statements (collecting 400 trajectories from both sets). Remarkably, models trained on synthetic data achieve nearly identical performance (27.8% PASS@1) to those trained on real data (28.0%). This finding validates the efficacy of our synthetic data generation methodology, demonstrating that procedurally generated environments can match the training value of real-world examples while providing scalability.

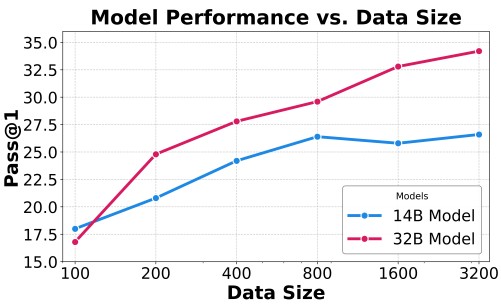

**Model Performance vs. Data Size**

Figure 2: **PASS@1 scaling curve with increasing number of training samples.** Performance improvement with more training samples, enabled by SWEGEN approach.

| Ablation | Config | PASS@1 (%) |
|---|---|---|
| Adding Thoughts | With | 34.4 |
| | Without | 30.4 |
| Real vs. Synthetic | Real | 28.0 |
| | Synthetic | 27.8 |

Figure 3: **Top.** Using thoughts in REACT agent trajectories leads to significant performance improvements. **Bottom.** Using SWEGEN synthetic generated issues and test cases achieves similar performance as real-world issues (400 trajectories for both real & synthetic in above) while providing better scalability during data collection.

**Explicit Thought Traces are Important.** During SFT we use both the agent's thought processes and actions as training targets. Models trained with thought demonstrations achieve significantly better performance compared to those trained without (34.2% vs 30.4% in Table 3). This suggests that exposing the model to step-by-step reasoning processes is necessary for reliable problem-solving in complex environments.

## 4 Efficient Inference Time Scaling With Hybrid Verifiers

We utilize R2E-Gym (§2) for inference-time scaling experiments with coding agents. In §4.1, we explore different axes for scaling test-time compute, focusing on two distinct approaches: 1) Execution-based Verifiers and 2) Execution-free Verifiers. We analyze the relative strengths and weaknesses of each approach, demonstrating their complementary nature (§4.2). Based on this insight, we propose a hybrid approach that leverages the strengths of both, significantly improving test-time performance (§4.3). Finally, we provide detailed ablations and analysis, examining critical design choices for our approach (§4.4).

### 4.1 Exploring Different Axes for Training Verifiers

Given an input task description $\mathcal{D}$, a set of agent trajectories $\{\mathcal{T}_i\}_{i=1}^K$ and candidate patch outputs $\{\mathcal{P}_i\}_{i=1}^K$, our objective is to build a verifier that assigns scores $\mathbf{S} = \{s_i\}_{i=1}^K$ to rank the outputs. To this end, we investigate two types of verifiers:

**Execution-Based Verifiers.** We train a specialized *testing-agent* that generates reproduction test cases to determine whether a candidate patch resolves the issue (i.e., whether the patch passes the generated test suite). Additionally, following Xia et al. (2024b), we leverage existing regression tests to filter out patches that fail to maintain backward compatibility. Our execution-based (EB) verifier thus comprises two components: 1) a *testing-agent* that generates targeted tests to evaluate bug fixes, and 2) a regression test filter that eliminates patches that compromise existing functionality. Specifically, we train the testing-agent (using QWEN-CODER-32B as base-model) to generate a comprehensive test script containing $M = 10$ diverse tests that cover various inputs, corner cases, *etc.*. See Appendix D for example generated tests. The execution-based score $s_k^{EB}$ for each each patch $\mathcal{P}_k$ is then computed as,

$$s_k^{EB} = \begin{cases} \text{TestScore}_k, & \text{if } RS_k = \max_{j \in [1,K]} RS_j, \\ 0, & \text{otherwise,} \end{cases} ; \text{ where } \quad \text{TestScore}_k = \sum_i \text{Pass}(\mathcal{P}_k, Test_i) \quad (1)$$

where $RS_k$ refers to the regression test score for the $k^{th}$ patch and helps selelct the patches with the highest regression test scores (Xia et al., 2024b). TestScore$_k$ is simply the sum of the number of passing tests for each patch $\mathcal{P}_k$. Please refer to Appendix §C for further details.

Notably, unlike zero-shot test generation with Agentless (Xia et al., 2024b), our testing agent interacts with the environment to examine existing test cases and generates new

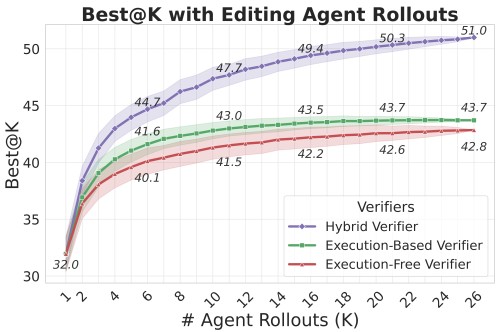
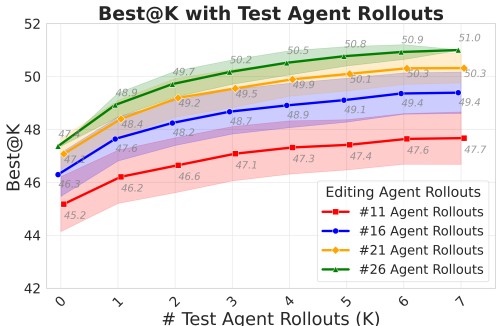

Figure 4: **Left.** BEST@K with increasing number of editing-agent rollouts. Inference-time scaling improves final performance for both execution-based and execution-free verifiers. Hybrid Verifier combining execution-based and execution-free verifiers provides significantly superior scaling. **Right.** BEST@K with increasing number of testing-agent rollouts. Increasing test-agent rollouts also improves final performance and can provide more compute efficient scaling than naively increasing only editing-agent rollouts.

Table 4: Performance of various models/methods on SWE-Bench Verified.

| Method | Model | Type | Verified |
|---|---|---|---|
| **Proprietary Models** | | | |
| Agentless-1.5 (Xia et al., 2024b) | GPT-4o | Pipeline | 34.0 |
| Agentless (Xia et al., 2024b) | O1 | Pipeline | 48.0 |
| Claude + Tools | Claude-3.6-Sonnet | Agent | 49.0 |
| Agentless-1.5 (Xia et al., 2024b) | Claude-3.6-Sonnet | Pipeline | 50.8 |
| OpenHands (Wang et al., 2024) | Claude-3.6-Sonnet | Agent | 53.0 |
| Claude + Tools | Claude-3.7-Sonnet | Agent | 62.3 |
| Claude + Tools (Best@Any) | Claude-3.7-Sonnet | Agent | 70.3 |
| **Open-source Models** | | | |
| SWE-SynInfer (Ma et al., 2024) | Lingma-SWE-GPT-72B | Agent | 30.2 |
| SWE-Fixer (Xie et al., 2025) | SWE-Fixer-72B | Pipeline | 30.2 |
| SWE-Gym (BEST@16 w/ Verifier) (Pan et al., 2024) | SWE-Gym-32B | Agent | 32.0 |
| SWE-RL (BEST@500 w/ Tests) (Wei et al., 2025) | SWE-RL-70B | Pipeline | 41.0 |
| Agentless (Xia et al., 2024b) | DeepSeek-R1 | Pipeline | 49.2 |
| **R2E-Gym (Ours)** (PASS@1) | R2E-Gym-32B | Agent | **34.4** |
| **R2E-Gym (Ours)** (BEST@16 w / Hybrid) | R2E-Gym-32B | Agent | **49.4** |
| **R2E-Gym (Ours) (BEST@26 w / Hybrid)** | R2E-Gym-32B | Agent | **51.0** |

tests informed by these examples with execution feedback. We demonstrate that this environment-aware approach provides additional benefits over zero-shot methods in §4.4.

**Execution-free Verifiers.** We next train execution-free (EF) verifiers for selecting the best trajectory from a set of sampled trajectories from the code-editing agent (§3). In particular, following (Pan et al., 2024), given task description $\mathcal{D}$, agent-trajectory $\mathcal{T}$ (sequence of thought, action, and observations) and output patch $\mathcal{P}$, we finetune a `Qwen2.5-Coder-14B` model to predict YES and NO tokens to determine correctness of a trajectory using SFT on correct and incorrect trajectories. The execution-free score is then computed by normalizing the relative probability of YES token as $s^{EF} = P(\text{YES})/(P(\text{YES}) + P(\text{NO}))$, where $P(\text{YES})$ and $P(\text{NO})$ are estimated through log-probabilities of corresponding token predictions.

## 4.2 Comparative Analysis of Execution-Based and Execution-Free Verifiers

**Experimental Methodology.** We evaluate verifier performance using the BEST@K metric, which quantifies each verifier's ability to identify correct patches from multiple candidates. Specifically, given $K$ trajectories, the BEST@K metric represents the percentage of problems where the verifier successfully selects the correct patch using its scoring mechanism. For our experiments, we sample 1 trajectory at temperature $T = 0$ and 25 trajectories at temperatures $T = 0.8$ and $T = 0.9$ from the R2E-Gym-32B model on SWEBENCH-VERIFIED problems. These trajectories achieve PASS@26 =64.4% (Figure 14). Next, we sample 7 tests using our testing

agent at temperature $T = 0.8$. When generating tests, the test agent is provided a *fixed* in-context example (from Django) showing sample starter code and format for writing test cases. We empirically find that use of an incontext example is useful for improving output formatting and lacking domain knowledge in the base LM; improving test generation for $\sim 2\%$ problems. Please see Listing C.1 for further details and incontext starter code.

**Both verifiers elicit inference time gains**. Figure 4 illustrates the BEST@K performance of both verifier types on the SWEBENCH-VERIFIED benchmark as a function of number of editing agent rollouts. Both execution-based and execution-free verifiers demonstrate substantial performance improvements with increased number of rollouts. However, BEST@K rate quickly plateaus for both methods, converging similarly to 43.7% and 42.8% respectively.

**Limited Distinguishability in Execution-Based Verifiers**. Recall that these verifiers output scores based on test pass counts and thus cannot differentiate between patches with identical test pass-rates, limiting their discriminative capacity. We study this discriminative capability from tests generated by our 32B testing agent, prompted SONNET-3.5-v2 model, and Agentless-1.5 reproduction tests (Xia et al., 2024b)[3] on a subset of SWEBENCH-VERIFIED problems. Figure 5 (left) presents the problem density distribution for distinguishability rate, i.e., the proportion of tests that successfully differentiate between top-ranked correct and incorrect patches. The results demonstrate that for the majority of problems, less than 20% of tests provide discriminative signal, constraining the re-ranking. Figure 6 additionally depicts that most generated tests either do not reproduce the bug (high Pass→Pass values in 6-left) or do not pass ground truth patches (high Fail→Fail values in 6-middle) primarily due to bugs or exceptions in the generated test cases.

**Vulnerability to Test Toxicity.** Following (Chen et al., 2022), we examine the prevalence of toxic tests, i.e., tests that pass incorrect patches but fail correct patches. Figure 5 (right) illustrates the distribution of toxic test rates across different test generation approaches. While toxic tests are generally rare, we find that for a small but significant subset of problems, testing agents generate toxic tests (up to 10% of total tests) that can erroneously rank incorrect patches above correct ones, undermining the reliability of execution-based verification.

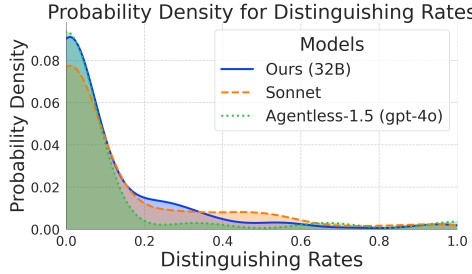 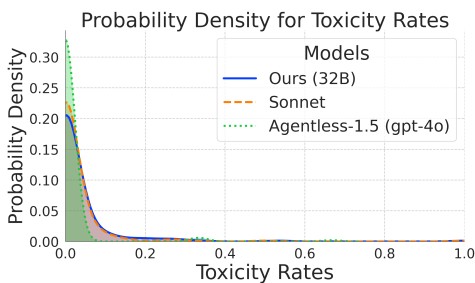

Figure 5: **Analyzing limitations of execution-based verifiers. Left:** Problem Probability Distributions for distinguishability rates depicting weak discrimination capabilities of tests. We observe that for the majority of problems, less than 20% of tests provide discriminative signal, constraining the re-ranking ability of test-based agent. **Right:** Distributions for toxicity rates showing (rare) generation of toxic tests. We find that execution-based verifiers are also vulnerable to (rare) generation of toxic tests (tests that pass incorrect patches but fail correct patches); which can undermine the reliability of execution-based verifiers.

**Execution-Free Verifiers can rely on heuristics**. We next study the workings and limitations of execution-free verifiers. In particular, we first perform quantitative ablation studies, studying the impact of different trajectory components (e.g., output patch, agent thoughts) to verifier performance. To this end, we train multiple execution-free verifiers (§4.1) excluding different trajectory components while training the verifier. Results are shown in Figure 7-a. We find that agent thoughts play a considerable role in determining the verifier performance. Surprisingly, the final BEST@26 drops from 42.8% to 37.6% when we remove the trajectory from the verifier input (i.e., only use the final patches). This means that while patch alone is

---

[3]We utilize test cases from the official artifacts repository (Xia et al., 2024a).

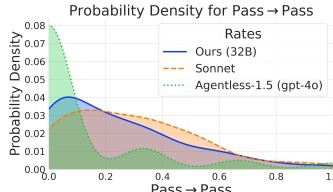 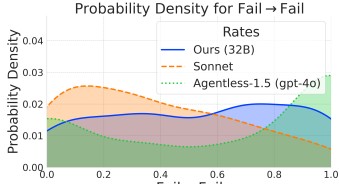 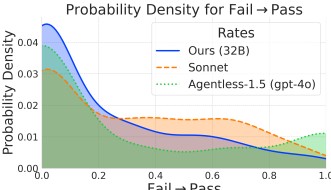

Figure 6: Problem Probability Distributions for Pass→Pass, Fail→Fail, and Fail→Pass generated test fractions for various approaches. We identify a large fraction of generated tests either do not reproduce the bug (left) or do not even pass the correct solution (middle).

| Method | Accuracy (%) | Best@26 (%) |
|---|---|---|
| Final Patch + Traj. | **71.82** | **42.8** |
| Patch Only | 68.01 | 37.6 |
| Traj. - Thoughts | 68.77 | 41.4 |

(a) **Impact of Patch & Thoughts** on execution-free verifier. Patch alone reduces performance, indicating that model relies on other heuristics (e.g., agent thoughts) for reranking; which can be misleading (see part-b: right).

1. **Successfully** reproduced the issue
2. Implemented a fix [...]
4. Ensured edge cases **are** handled
5. **Maintained** backward compatibility [...]
<function=finish>submit</function> [...]

**Great**! The fix **works**. Let's see what we did to fix the issue:
1. We identified that the original code was failing because it was trying to use the `.inverse()` method directly on permutations, which [...]

(b) Top two attention windows while predicting YES for an incorrect trajectory. We find that focusing on heuristics (agent thoughts) can mislead the verifier.

Figure 7: **Quantitative and qualitative analysis on limitations of execution-free verifiers.** We perform two experiments: a) Quantitative ablations on the impact of output patch on verifier performance; showing that execution-based verifiers rely on other heuristics (e.g., agent thoughts) over the final patch. b) Qualitative visualization analyzing top $k = 2$ sliding windows with highest mean attention score while predicting output token YES (§4.2) for an *incorrect* agent trajectory (sympy__sympy-24443: SWE-Bench (Yang et al., 2024b)). Focusing on heuristics (*e.g.*, agent thoughts) can be misleading, and the verifier predicts the trajectory as correct. Visualizations are condensed for space. Please refer to the Appendix for further visualizations and results.

responsible for determining the correctness, execution-free verifiers heavily rely on trajectory features, such as agent thoughts, to make predictions.

To further investigate this phenomenon, we also perform an attention analysis trying to visualize parts of the input trajectory which are most relevant while predicting the output success with execution-free verifiers. In particular, we perform a sliding window search over the input trajectory, and compute the mean attention score over the tokens in the window when predicting the final output token (YES: correct, NO: incorrect). Figure 7 (right) illustrates the top two windows receiving the highest attention scores, demonstrating that verifiers disproportionately attend to agent thoughts. This can be misleading since the verifier can use these sentiment signals in these thoughts as proxies for correctness rather than evaluating the technical merits of the solution (i.e. the output patch).

### 4.3 Hybrid Inference Time Scaling

**Combining the verifier strengths.** Given the analysis from §4.2, we can summarize two key insights: 1) Execution-based approach provides direct signal for patch correctness through execution but suffers from lack of distinguishing tests 2) Execution-free approach offers better distinguishability between patches through a continuous reward score $s^{EF}$ but can be biased to pay more attention to heuristics (e.g., agent thoughts) over final output patch.

Given the above insights, we thus propose a hybrid verifier that leverages the strengths of both approaches. Particularly, we define the hybrid verifier with score $s_k^H$ as,

$$s_k^H = \text{Top}_n(s_k^{EF}) + s_k^{EB}, \text{ where } \text{Top}_n(s_k^{EF}) = \begin{cases} s_k^{EF}, & \text{if } s_k^{EF} \text{ is among the top } n \text{ scores,} \\ -\infty, & \text{otherwise.} \end{cases} \quad (2)$$

where $s_k^{EB}$ provides execution-feedback, $s_k^{EF}$ provides distinguishability in case of a tie with execution-based test scores (as $s_k^{EF}$ provides a continuous score between 0 and 1), and $\text{Top}_n$ restricts hybrid verifier to only consider the top verifier ranked patches. In practice, we perform regression filtering after the top-n filtering to ensure non-zero scores.

**Main Results.** Results are shown in Tab. 4 and Fig. 4. While both execution-based and execution-free methods rapidly reach performance plateaus with increasing agent rollouts (saturating at $\sim$ 43%), our hybrid approach demonstrates substantially superior scaling properties, yielding significant performance improvements (additional 7-8%); achieving a BEST@26 performance of 51% on the challenging SWEBENCH-VERIFIED benchmark.

**Comparison to Open Systems**. The proposed approach significantly outperforms other open-weight alternatives; reflecting a new state-of-the-art in this domain. Among other generalist-agent methods, SWE-Gym (Pan et al., 2024) recently achieves a BEST@16 performance of 32.0%. Similarly, concurrent work (Wei et al., 2025) recently achieved 41.0% using RL and BEST@500 (using Agentless). In contrast, despite mainly relying on supervised fine-tuning for training, our proposed approach achieves a PASS@1 itself of 34.4% with BEST@26 performance of 51.0% — achieving strong performance improvements through simply more scalable data curation (§2) and better test-time scaling (Figure 4).

### 4.4 Ablation Studies on Hybrid Verification Design

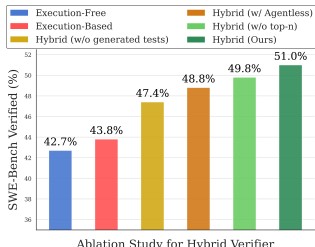

Figure 8: **Ablation Study on Hybrid Verifier.** We find three key insights: 1) While both execution-based and execution-free verifiers saturate around 42-43%, the hybrid approach yields significantly higher test-time gains (51%). 2) Regression tests alone are insufficient for hybrid scaling — achieving only 47.4% aggregation performance. 3) Agentic vs Agentless: training a specialized testing agent is important improving the performance from 48.8% to 51%.

**Variation with Test-Agent Rollouts.** As in 4.2, execution-based test generation can suffer from a lack of distinguishing tests. One approach to address this, is to sample more test-agent rollouts. We quantify this effect in Figure 4 (right). We observe that increasing number of test-agent rollouts consistently helps improve performance with our hybrid approach.

**Compute-Efficient Rollouts.** Figure 4 (right) illustrates the BEST@K performance as a function of both test-agent and code-editing agent rollout counts. Interestingly, we find that sampling more test-agent rollouts can provide more compute optimized inference-scaling over naively sampling more editing-agent rollouts. For instance, increasing the number of editing-agent rollouts from 16 to 21 improves the BEST@K performance from 47.6% to 48.4%. In contrast, simply sampling 5 more test-rollouts can yield better gains (BEST@K 49.3%).[4]

**Regression Tests Alone are Insufficient.** Our execution-based verification framework integrates both regression and generated reproduction tests. Figure 5 (right) isolates the impact of regression tests alone on the final performance. While regression tests alone improve performance from 42.9% to 47.4%, using generated tests further enhances performance to 51.0%, demonstrating that both test types provide essential and complimentary signals.

**Agentic vs Agentless Tests.** A distinguishing feature of our approach is to train a specialized agent for test-generation; instead of the zero-shot approach from Xia et al. (2024b). To evaluate this design choice, we conducted a controlled comparison using official Agentless tests from their released artifact (Xia et al., 2024a) within our hybrid verification framework on the SWEBENCH-VERIFIED benchmark. Figure 5 (right) demonstrates that while Agentless tests provide meaningful performance improvements, our agent-generated tests yield superior results (51.0% versus 48.8%), validating our agent-based approach to test generation.

---

[4]Note that test-agent rollouts are also usually considerably cheaper than editing-agent rollouts.

**Role of** $\text{Top}_n$. We evaluate the impact of the $\text{Top}_n$ filtering mechanism introduced in Equation (2). Figure 5 (right) shows that this selective application strategy improves performance from 49.8% to 51.0%. This improvement likely stems from mitigating the impact of toxic tests (§4.2) by restricting their application to higher-quality patches (identified via execution-free reward scores $s_k^{EF}$), thereby enhancing the reliability of the verification process.

# 5 Related Work

**Programming Agents**. Recent work on GITHUB issue resolution includes SWE-agent (Yang et al., 2024b), Autocoderover (Zhang et al., 2024b), OpenHands (Wang et al., 2024), Agent-Less (Xia et al., 2024b), Moatless Orwall (2024). All of them rely on proprietary models due to a lack of datasets and open-weight models —— a gap our work addresses.

**Agent Training Environments**. Existing SWE agent environments have key limitations: SWE-Bench (Jimenez et al., 2023) lacks executable training environments, R2E (Jain et al., 2024b) offers only 246 instances with function completion. SWE-Gym (Pan et al., 2024) collects executable GITHUB environments similar to us but rely on human-written issues and test cases. Synthetic data generation has been studied in various domains but our work is the first to apply it for executable GITHUB environment collection. We use back-translation (Li et al., 2024) and test-generation in SWEGEN approach. Please see Long et al. (2024) for a comprehensive survey on synthetic data generation methods.

**SWE-Agent Training**. Ma et al. (2024) and Xie et al. (2025) train on synthetic code editing tasks. Pan et al. (2024) study SFT on agent trajectories and inference scaling similar to our work. Wei et al. (2025) explores reinforcement learning on large scale data collected from real-world GITHUB issues without execution feedback.

**Verifiers for SWE-Coding Tasks**. Various works have explored use of verifiers for SWE tasks. AgentLess (Xia et al., 2024b) used majority voting to select the best patch from multiple agents. Agentless-1.5 relied on reproduction and regression tests to verify the correctness of generated patches. Zhang et al. (2024a) proposed multi-agent commitee-review (LLM judge) to select the best patch from multiple agents. Pan et al. (2024) proposed trajectory verifiers to re-rank the generated patches based on LLM score.

**Verifiers for General Coding Tasks**. Various works have explored the use of verifiers for general coding tasks on isolated puzzles (HumanEval (Chen et al., 2021)), interviews (Jain et al., 2024a), and competition or olympiad problems (Hendrycks et al., 2021; Li et al., 2022) Gu et al. (2024) showed that LLM judges perform poorly on checking correctness of generated code. Chen et al. (2022); Ridnik et al. (2024); Key et al. (2022); Zhang et al. (2023a) study how test generation can be used to re-rank the generated code samples. Inala et al. (2022); Zhang et al. (2023b); Ni et al. (2023) employ neural code re-ranker models.

In this work, we extend these lines of work by first presenting **novel insights on challenges and opportunities for both execution-based and execution-free approaches in SWE-Coding. Using these insights, we also propose a novel hybrid approach that effectively combines their strengths** to achieve better performance (51.0% on SWEBENCH-VERIFIED).

# 6 Conclusion

In this paper, we introduce R2E-Gym, the largest gym environment and training framework for scaling open-weight SWE agents. We share two key insights: 1) Synthetic data curation can enable more scalable training on SWE tasks. 2) Hybrid-test time scaling: different axis for test-time scaling (execution-based testing agents and execution-free verifiers) exhibit complementary strengths; which can be leveraged to achieve significantly higher test-time gains. Overall, our final approach achieves 51% on SWE-Bench Verified, reflecting a new state-of-the-art for open-weight SWE agents, while also for first-time showing competitive performance with some proprietary models. We hope that our work can offer unique insights for scaling open-source SWE-agents on real-world applications.

## Acknowledgement

N. Jain and M. Shetty are supported by NSF grants CCF:1900968, CCF:1908870, and by SKY Lab industrial sponsors and affiliates. This work is additionally supported by the R2E OpenPhilanthropy grant.

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

# A  Dataset Details

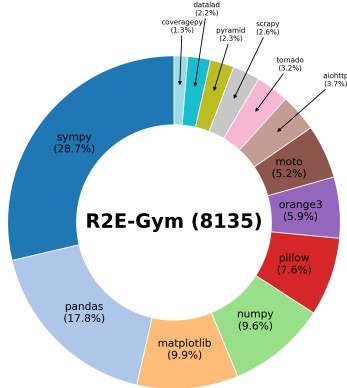

Figure 9: Repo distribution for our complete R2E-Gym dataset consisting of 8135 instances.

**Commit Filtering Heuristics.** Our commit filtering approach employs multiple heuristics to identify high-quality bug fixes and improvements suitable for training data. We particularly filter for small scoped changes, prioritizing non-documentation updates, and correlated code and test matches. We perform this filter at both line and AST entity level. To ensure consistency and quality, we employ specific thresholds in our filtering process:

- Maximum of 5 non-test files modified in a single commit
- Maximum of 100 edited lines across all non-test files
- Maximum patch length of 2000 characters to ensure focused changes
- No more than 1 deleted entity in non-test files
- Maximum of 3 added entities in non-test files
- Maximum of 3 edited entities in non-test files
- No more than 10 statement-level changes to maintain tractability

Additionally, we use LLM as a judge filter to further refine our dataset.

**Repository Installation.** Installing historical commits from GitHub repositories presents significant challenges due to evolving dependency requirements and API changes. We use a Docker-based approach with a search-based dependency resolution strategy to create reproducible environments for each commit. Our installation process follows these steps:

1. Extract dependency information from `requirements.txt, setup.py,` etc
2. Iteratively identify potential version conflicts and compatibility issues
3. Generate multiple candidate dependency configurations
4. Test each configuration until a working environment is found

This process is semi-manual and challenging to scale and we aim to rely more on LLMs in the future. Example installation scripts test multiple dependency combinations sequentially, exiting on the first successful build:

```
build_and_check_pandas(){
    local python_version=$1;
    local numpy_version=$1;
    local setuptools_version=$3$
    ...
```

```
}

# Attempt with first configuration
if build_and_check_pandas "3.7" "1.17.*" "<0.30" "62.*" "0.23"; then
  echo "[INFO]␣First␣combo␣succeeded.␣Exiting."
  exit 0
fi

# Attempt with second configuration
if build_and_check_pandas "3.8" "1.20.*" "<0.30" "62.*" "0.23"; then
  echo "[INFO]␣Second␣combo␣succeeded.␣Exiting."
  exit 0
fi

# Attempt with third configuration
if build_and_check_pandas "3.10" "1.26.*" "===3.0.5" "62.*" "0.23"; then
  echo "[INFO]␣Third␣combo␣succeeded.␣Exiting."
  exit 0
fi
```

Listing 1: Example installation script excerpt

This approach allows us to create working environments for historical commits, enabling execution-based validation of our dataset.

**Test Generation.** We use an Agentless-like reproduction test generation approach. A key difference is that we use the ground truth patch as context when generating the tests.

**Issue Generation.** As discussed in the main paper, we use backtranslation to generate synthetic issues for commits that lack human-written GitHub issues. Our approach leverages both the code changes in the commit and the test execution results to create realistic, informative issue descriptions. The issue generation process follows these steps:

1. Extract failing test functions from the execution results
2. Analyze test outputs to identify error messages and expected behaviors
3. Provide the LLM with commit message, code patch, and test execution results
4. Guide the LLM to generate a concise, informative issue that describes the bug without revealing the solution

For each commit, we extract and utilize specific components:

- **Commit metadata**: Hash and commit message provide context about the change
- **Code patches**: We separate non-test file changes (showing what was fixed) from test file changes (showing how to verify the fix)
- **Test execution**: We include both old (failing) and new (passing) executions
- **Test functions**: We extract relevant test functions that demonstrate the bug
- **Assertion failures**: We extract and format the failing assertions from the old commit to show error details

The prompt construction carefully organizes these components to give the LLM sufficient context while focusing attention on the most relevant information for issue generation. We carefully design our prompting strategy to ensure the generated issues resemble human-written ones, focusing on clarity, naturalness, and providing sufficient information for understanding the bug.

```
# Build the complete prompt with all components
def get_prompt(commit, execution_result, issues=None):
    # Include commit hash and message
    # Include commit patch (non-test files)
    # Include test file changes
```

```
    # Include execution results from old and new commits
    # Include improved test functions
    # Include test function code
    # Include assertion failures
    # Include example issues and instructions
```

Listing 2: Issue generation code structure

The template below shows our prompt guidelines:

```
As you are trying to generate synthetic issues, you will follow these
    guidelines:

1. Keep the issue concise and informative.
2. Describe the failing test, including the input that causes the failure
    , the nature of the failure, and the expected behavior. Do NOT
    mention test functions or files directly.
3. Do not reveal the solution to the problem in the issue. Only describe
    the bug and the expected behavior.
4. If there are multiple failing tests, focus on the most informative one
    or a subset that best describes the general nature of the failure.
5. Describe the expected output of the failing test:
   - For errors, describe the error message.
   - For failing tests, mention what is supposed to happen.
6. Write the issue as a human would, using simple language without
    excessive formatting.
7. Use concrete terms to describe the nature of the failure. Avoid vague
    terms like "specific output" or "certain data".
8. INCLUDE test code to describe the bug but keep it brief and relevant.
    Truncate or simplify tests longer than 5-6 lines.
9. Do not mention external files unless absolutely necessary.
10. Format code snippets using triple backticks.

The issue should include:
1. A clear and concise title
2. A description of the problem with detailed example buggy code
3. Expected behavior
4. Actual behavior or error message
```

Listing 3: Issue generation template

This approach enables us to generate high-quality synthetic issues that provide clear problem statements for our training data, even for commits that lack human-written issues. Below are examples of synthetic issues generated using our approach:

```
**Title:** Calling `load()` Before `draft()` Causes `draft()` to Fail for
    JPEG Images

**Description:**
When generating a thumbnail for a JPEG image using the `thumbnail()`
    method, the method calls `load()` before `draft()`. This sequence
    results in the `draft()` method returning `None`, which prevents the
    thumbnail from being properly optimized.

**Example Code:**
```python
from PIL import Image

with Image.open("Tests/images/hopper.jpg") as im:
    im.thumbnail((64, 64))
```

**Expected Behavior:**
```

```
The `thumbnail()` method should utilize the `draft()` method to optimize
    the image size before loading, ensuring that the thumbnail is resized
     correctly and efficiently.

**Actual Behavior:**
The `draft()` method returns `None` because `load()` is invoked before it
    . This prevents the thumbnail from being optimized, potentially
    leading to incorrect thumbnail sizes or unnecessary memory usage.
```

Listing 4: Example synthetic issue for a PIL image thumbnail bug

```
**Title:** Unable to Register Route with Names Containing Both Dots and
    Colons

**Description:**
After merging branch '0.18', attempting to register a route with a name
    that includes both dots (`.`) and colons (`:`) results in a `
    ValueError`. The recent changes were intended to allow route names to
     be a sequence of Python identifiers separated by dots or colons, but
     this combination is still causing issues.

**Example Code:**
```python
from aiohttp.web import UrlDispatcher, PlainRoute

def handler(request):
    return 'Hello'

router = UrlDispatcher()

# Attempting to register a route with both dots and colons in the name
route = PlainRoute('GET', handler, 'test.test:test', '/handler/to/path')
router.register_route(route)
```

**Expected Behavior:**
Registering a route with a name like `'test.test:test'` should succeed
    without errors, as the name follows the updated rules allowing
    multiple identifiers separated by dots or colons.

**Actual Behavior:**
A `ValueError` is raised with the message:
```
ValueError: Incorrect route name value, Route name should be a sequence
    of python identifiers separated by dot or column
```
This prevents the registration of route names that include both dots and
    colons, contrary to the intended flexibility introduced in the recent
     commit.
```

Listing 5: Example synthetic issue for a route name validation bug

**Patch Minimization.** We identify that the ground-truth patches often contain irrelevant code changes that are not required to fix the bug, often making modifications to style and structure of the programs. We implement a patch-minimization approach to identify the minimal set of code changes required to fix the bug by iteratively removing the code changes and checking whether the tests still pass. This allows us to collect fine-grained signal for evaluating localization capabilities of LLMs.

## B  SFT Training

**Agent Details.**

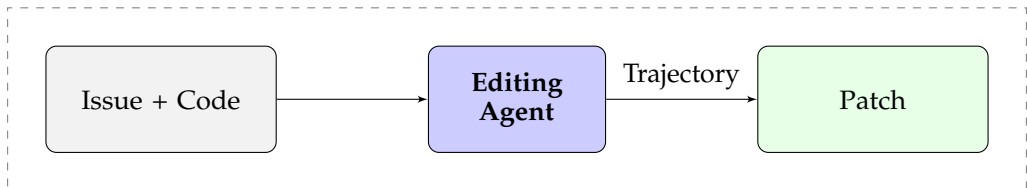

Figure 10: Code-editing agent architecture: The agent takes an issue description and codebase as input and produces a patch that fixes the issue.

We use R2E-Gym to train a general-purpose prompting agent. In particular, we train our code-editing agent on tasks from R2E-Gym, where given an executable environment $\mathcal{E}$ and problem description $\mathcal{D}$, the agent is asked to solve the provided issue using any means necessary. Particularly, unlike (Orwall, 2024), we do not rely on the use of specialized workflows. The agent is tasked to solve the entire task end-to-end, including writing its own reproduction scripts, finding the bug, proposing a fix and then testing its correctness. Similar to (Wang et al., 2024), the agent is also provided with a finish tool, allowing it to submit a solution if it thinks it has completed the task.

**Agent and Tools.** Similar to (Aleithan et al., 2024; Wang et al., 2024), we adopt the traditional REACT format (Yao et al., 2022) for agent-design. For AGENTHUB, we use a minimalistic set of four tools to enable the agent to perform diverse SWE tasks; 1) `file_editor:` for viewing and editing files, 2) `search_tool:` for searching a relevant term in a given file or folder, 3) `execute_bash:` allowing execution of non-interactive bash commands (*e.g.*, for running test scripts), 4) `submit:` for ending the current trajectory while returning expected outputs. No internet or browser access is provided to the agent during the training process.

**Data Curation.** For training, we use supervised finetuning with rejection sampling using trajectories from `sonnet-3.5` model for supervision. To avoid contamination, we only use a subset of R2E-Gym consisting of repos with no overlap with the SWE-Benchdataset. The resulting subset (R2E-Gym-lite) consists of 4538 executable environments across 10 repositories (Figure 2). Overall, we collect a total of 3321 successful trajectories from 2048 unique test environments. For rejection sampling we use the unit tests from R2E-Gym environments (both synthetic and existing). For each trajectory, we use a maximum of $N = 40$ steps. Also, we limit the number of tokens per-trajectory to 32K max tokens. Finally, we also use a maximum timeout of 10-min for the overall trajectory and 90 seconds for each action execution, in order to avoid cases where the agent launches a long-running background process. We collect all training data using a temperature of 0.2.

**Training Setup and Hyperparameters.** For training, we use the `Qwen-2.5-Coder` 7B, 14B and 32B series as the base model for training SWE-agents on R2E-Gym. For training we perform full SFT using the above collected trajectories using LLaMA-Factory (Zheng et al., 2024). We train the overall model for a total of 2 epochs, batch size as 8 while using a learning rate of $1e^{-5}$. The warmup ratio for training was set to 0.1. Due to computational constraints, a maximum context length of 20K was used for training the agent. In future, the use of context-parallelism can enable us to further push the performance when training SWE-agents on more complex tasks requiring larger-context lengths.

## C    Inference Time Scaling

### C.1    Execution-Based Testing Agents

**Agent Details.** We train a specialized *testing-agent* that generates reproduction test cases to determine whether a candidate patch resolves the issue (i.e., whether the patch passes the generated test suite). Specifically, we train the testing-agent (using QWEN-CODER-32B as base-model) to generate a comprehensive test script containing $M = 10$ diverse tests that cover various inputs, corner cases, etc. We use the same agent scaffold from Sec. 3 for training the testing agent.

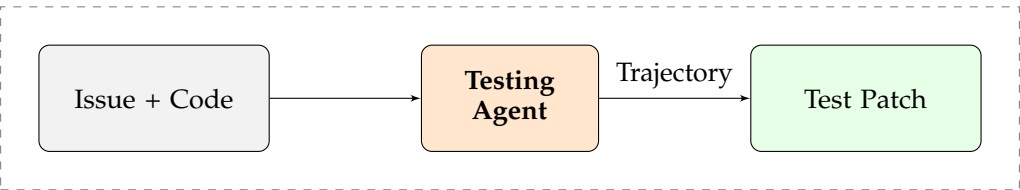

Figure 11: Testing agent architecture: The agent generates comprehensive test cases to verify if a candidate patch resolves the issue.

**Data Curation.** For training, we use supervised finetuning using trajectories from sonnet-3.5 model for supervision. Overall, we collect a total of 2203 test-generation trajectories from sonnet (both positive and negative trajectories with minimal rejection sampling). For each trajectory, we use a maximum of $N = 40$ steps. Also, we limit the number of tokens per-trajectory to 20K max tokens. Finally, we also use a maximum timeout of 5-min for the overall trajectory and 60 seconds for each action execution, in order to avoid cases where the agent launches a long-running background process.

**Training Setup and Hyperparameters.** For training, we use the QWEN-CODER-32B model as the base model. We then use the above collected training SFT trajectories to perform full finetuning with the QWEN-CODER-32B model using LLaMA-Factory (Zheng et al., 2024). We train the overall model for a total of 2 epochs, batch size as 8 while using a learning rate of $1e - 5$. A maximum context length of 20K was used for training the agent. The warmup ratio for training was set to 0.1.

**In-Context Starter Code Demonstration**. We provide the following in-context starter-code demonstration (from the Django repository) to the testing agent.

```python
import os
import django
from django.conf import settings
from django.db import models
from django.test import TestCase
from django.test.utils import setup_test_environment

# Configure Django settings before setup
os.environ.setdefault('DJANGO_SETTINGS_MODULE', 'tests.test_sqlite')

# Override settings
settings.configure(
    DATABASES={
        "default": {
            "ENGINE": "django.db.backends.sqlite3",
            "NAME": "test.db",
            "TEST": {
                "NAME": "test.db",
            },
        }
    },
    INSTALLED_APPS=["tests"],
    MIGRATION_MODULES={"tests": None},  # Disable migrations for the
        tests app
)

# Setup Django
django.setup()
setup_test_environment()

# Define test models
class ExampleModel(models.Model):
    example_char = models.CharField(max_length=255)
    example_int = models.IntegerField()
```

```python
    class Meta:
        app_label = 'tests'  # Set the app_label to 'tests'

# Create the database tables
from django.core.management import call_command
call_command('migrate', run_syncdb=True)

def add_test_data():
    """Create test instances of the model"""
    ExampleModel.objects.create(example_char="Test_1", example_int=1)
    ExampleModel.objects.create(example_char="Test_2", example_int=2)

# Add test data
add_test_data()
```

Listing 6: Incontext Demonstration for Testing Agent

## C.2 Execution-Free Verifiers

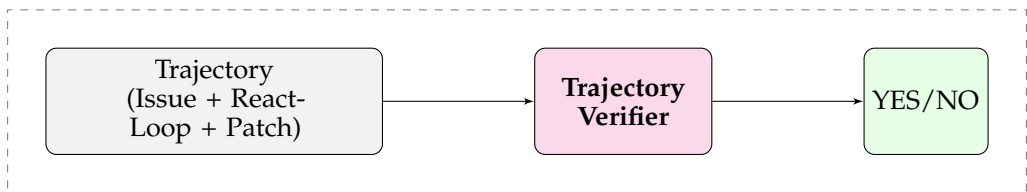

Figure 12: Execution-free verifier architecture: The verifier predicts whether a patch is correct based on the full trajectory without executing the code.

**Verifier Details.** In addition to the execution-based "testing agents", we also explore the execution-free outcome-supervised reward models (a.k.a verifiers) (Cobbe et al., 2021). In particular, given a problem statement $\mathcal{D}$, agent-trajectory $\mathcal{T} = \{a_1, o_1, a_2, o_2, \ldots, a_n, o_n\}$ and output patch $\mathcal{O}$ from the code-editing agent on the R2E-Gym environments, we train a `Qwen2.5-Coder-14B` model (Yang et al., 2024a) to output a scalar score value $s^{EF} \in [0, 1]$ predicting the probability of output patch being correct. Specifically, following (Pan et al., 2024) we output the correctness of each patch through output tokens YES (correct) and NO (incorrect). The overall reward score is then computed by normalizing the relative probability of YES token as $r = P(\text{YES})/(P(\text{YES}) + P(\text{NO}))$, where $P(\text{YES})$ and $P(\text{NO})$ are estimated through the log-probabilities of the corresponding token predictions.

**Training Data.** We first use the trajectories collected for code-editing agent training §3 in order to obtain a collection of positive and negative samples for verifier training. Following the best configuration from (Pan et al., 2024), we also generate on-policy trajectories using our trained 32B model. We then filter the collected samples to have an equal number of positive and negative samples. The overall dataset consists of 5700 total trajectories including both positive and negative samples. For training, we follow the template from (Pan et al., 2024), asking the LLM model to predict the output as YES for positive and NO for negative trajectories.

**Training Setup and Hyperparameters.** For training, we use the QWEN-CODER-14B model as the base model. We then use the above collected training SFT trajectories to perform finetuning using LLaMA-Factory (Zheng et al., 2024). Similar to (Pan et al., 2024), we perform LORA finetuning using a rank of 64. We train the overall model for a total of 2 epochs, batch size of 8 while using a learning rate of $1e - 5$. A maximum context length of 32K was used for training the agent. The warmup ratio for training was set to 0.1.

### C.3 Execution-Based Analysis

In our analysis of execution-based testing agents, we focus on two key metrics: distinguishability and toxicity of generated tests. These metrics help us understand the effectiveness and limitations of execution-based verification.

**Distinguishability Rate.** The distinguishability rate measures a test's ability to differentiate between correct and incorrect patches. A test is considered "distinguishing" if it behaves differently when applied to correct patches versus incorrect patches. In practical terms, this means the test can help us identify which patches are correct and which are not.

For example, consider a test that passes for all correct patches but fails for all incorrect patches—this test has perfect distinguishability. Conversely, a test that passes (or fails) for both correct and incorrect patches provides no useful signal for distinguishing between them. Mathematically, for a given test $t$ and a set of patches $P$ divided into correct patches $P_c$ and incorrect patches $P_i$, we compute distinguishability metric as:

$$\text{Distinguish}(t) = \mathbb{1}\left[\max_{p \in P_i} \text{Pass}(p, t) \neq \max_{p \in P_c} \text{Pass}(p, t)\right] \tag{3}$$

where $\text{Pass}(p, t)$ indicates whether patch $p$ passes test $t$, and $\mathbb{1}[\cdot]$ is the indicator function. This formula checks whether the best-performing incorrect patch behaves differently on the test compared to the best-performing correct patch. The distinguishability rate for a set of tests $T$ is then the average distinguishability across all tests:

$$\text{DistinguishRate}(T) = \frac{1}{|T|} \sum_{t \in T} \text{Distinguish}(t) \tag{4}$$

In our analysis, we found that most generated tests have low distinguishability rates—typically less than 20% of tests can effectively differentiate between correct and incorrect patches. This limitation significantly impacts the ability of execution-based verification to identify the best patches, especially as the number of candidate patches increases.

**Toxicity Rate.** We define toxic tests as those that incorrectly favor incorrect patches over correct ones. The toxicity rate is the proportion of tests that exhibit this behavior. Mathematically:

$$\text{Toxic}(t) = \mathbb{1}\left[\max_{p \in P_i} \text{Pass}(p, t) > \max_{p \in P_c} \text{Pass}(p, t)\right] \tag{5}$$

The toxicity rate for a set of tests $T$ is:

$$\text{ToxicityRate}(T) = \frac{1}{|T|} \sum_{t \in T} \text{Toxic}(t) \tag{6}$$

While toxic tests are generally rare, they can significantly impact verification reliability when present, with toxicity rates reaching up to 10% for some problems. These findings highlight the importance of generating diverse, high-quality tests and the value of combining execution-based verification with other approaches, such as execution-free verifiers, to achieve more robust results.

### C.4 Execution-Free Analysis

Figure 13 shows the limitations of the execution-free verifier.

## D Example Testing Agent Outputs

This section provides examples of test cases generated by our approach.

```
Since we've:
 1. Successfully reproduced the issue
 2. Implemented a fix
 3. Verified the fix works
 4. Ensured edge cases are handled
 5. Maintained backward compatibility

I believe we can now submit this as the
    solution:

<function=finish>
  <parameter=command>submit</parameter>
  <parameter=result>Fixed the `
      _check_homomorphism` function...
```

```
[ASSISTANT]
Great! The fix works. Let's analyze what we
    did:

1. We identified the original code was failing
    because
   it was using a `.inverse()` method directly
   , which
   doesn't exist for permutations.
2. We fixed the issue by:
   - Using `**-1` for inverses, which works
      for both
     Permutations and FreeGroupElements
```

```
[ASSISTANT]
The changes look good. I've made the following
    improvements:
1. Simplified logic to check both the
   generator and inverse
2. Used the `**-1` operator for Permutation
   and FreeGroupElement
3. Maintained robust handling of inverted
   generators

Let's run the reproduce script again:

<function=execute_bash>
  <parameter=cmd>python reproduce_issue.py</
      parameter>
</function>
[/ASSISTANT]
```

```
[USER]
Exit code: 0
Execution output of [execute_bash]:
[STDOUT]

Testing with DihedralGroup(3):
SUCCESS: Homomorphism created successfully

Testing with SymmetricGroup(3):
SUCCESS: Homomorphism created successfully
```

Figure 13: **Analyzing limitations of execution-free verifiers**. Qualitative visualization analyzing top $k = 4$ sliding windows over agent trajectory (refer §4.2) with highest mean attention score (sympy__sympy-24443: SWE-Bench). We observe that the RM can be biased by the agent thought / actions, instead of relying on the final output patch.

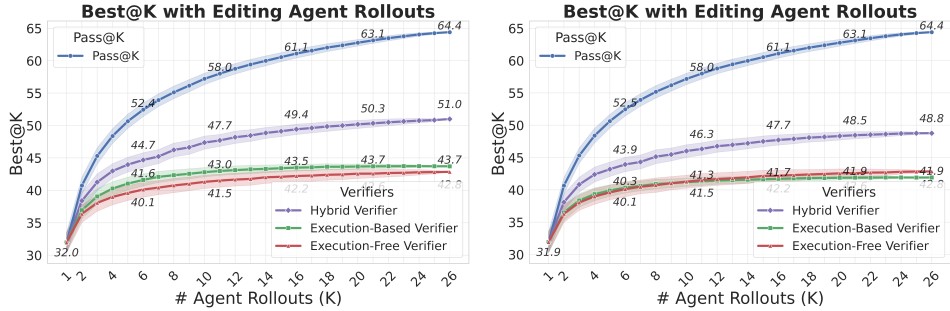

Figure 14: Pass@K plot for our agent and using Agentless tests respectively.

### D.1 Example 1: SymPy Relational Parsing Tests

The following example shows a truncated test suite for validating relational parsing in SymPy, demonstrating our approach's ability to generate multiple test cases. This test was generated to address the issue in SymPy PR #24661, which fixes relational parsing in the SymPy library.

```python
from sympy import Lt, Gt, Le, Ge, Eq, Ne

def test_relational_parsing():
    # Test case 1: Basic less than operation
    try:
        result = parse_expr('1 < 2', evaluate=False)
        expected = Lt(1, 2, evaluate=False)
        if str(result) == str(expected):
            print("Test Case 1: Issue resolved")
        else:
```

```
            print("Test Case 1: Issue reproduced")
    except Exception as e:
        print("Test Case 1: Other issues")

    # Test case 2: Greater than operation
    try:
        result = parse_expr('3 > 2', evaluate=False)
        expected = Gt(3, 2, evaluate=False)
        if str(result) == str(expected):
            print("Test Case 2: Issue resolved")
        else:
            print("Test Case 2: Issue reproduced")
    except Exception as e:
        print("Test Case 2: Other issues")

    # ... [6 more test cases omitted for brevity] ...

    # Test case 9: Chained comparisons
    try:
        result = parse_expr('1 < x < 2', evaluate=False)
        if isinstance(result, bool):
            print("Test Case 9: Issue reproduced")
        else:
            print("Test Case 9: Issue resolved")
    except Exception as e:
        print("Test Case 9: Other issues")
```

Listing 7: Test cases for SymPy relational parsing (truncated). Successfully detects incorrect code from correct code.

### D.2 Example 2: Django Model Choice Field Validation Tests

The following truncated example demonstrates tests for Django's ModelChoiceField validation with proper setup and teardown code. This test was generated to address the issue in Django PR #13933, which improves error messages in ModelChoiceField validation.

```
from django.forms import ModelChoiceField, ModelMultipleChoiceField
from django.db import models
from django.core.exceptions import ValidationError
from django.test import TestCase
from django.db import connection

# Create a simple test model
class TestModel(models.Model):
    name = models.CharField(max_length=100)

    def __str__(self):
        return self.name

    class Meta:
        app_label = 'test_app'

def setup_test_environment():
    # Create the test table
    with connection.schema_editor() as schema_editor:
        schema_editor.create_model(TestModel)

    # Create some test data
    TestModel.objects.create(name="Option 1")
    TestModel.objects.create(name="Option 2")
    TestModel.objects.create(name="Option 3")

def cleanup_test_environment():
```

```python
    # Drop the test table
    with connection.schema_editor() as schema_editor:
        schema_editor.delete_model(TestModel)

def test_modelchoice_validation():
    try:
        # Test Case 1: Basic invalid choice for ModelChoiceField
        field = ModelChoiceField(queryset=TestModel.objects.all())
        try:
            field.clean(999)  # Non-existent ID
            print("Test Case 1: Issue resolved")
        except ValidationError as e:
            if "999" not in str(e):  # Value should be in error message
                print("Test Case 1: Issue reproduced")
            else:
                print("Test Case 1: Issue resolved")

        # ... [4 more test cases omitted for brevity] ...

        # Test Case 6: Valid choice
        obj = TestModel.objects.first()
        try:
            result = field.clean(obj.id)
            if result == obj:
                print("Test Case 6: Issue resolved")
            else:
                print("Test Case 6: Issue reproduced")
        except ValidationError:
            print("Test Case 6: Issue reproduced")
    except Exception as e:
        print(f"Unexpected error: {e}")
```

Listing 8: Test cases for Django ModelChoiceField validation (truncated). Most test cases error due to unhandled exceptions and do not distinguish.

## E   Agent Trajectory Example

This section provides a visual example of an agent's trajectory while solving a software engineering task. The sequence shows the step-by-step process from problem statement to solution, demonstrating how our agent approaches and solves real-world programming issues.

This trajectory example illustrates several key aspects of our agent's problem-solving approach:

- **Systematic Exploration:** The agent methodically explores the codebase to understand the context and locate the issue.
- **Root Cause Analysis:** Rather than addressing symptoms, the agent identifies the underlying cause of the problem using test_issue.py.
- **Solution Development:** The agent formulates a clear plan before implementing changes.

These capabilities enable our agent to effectively tackle complex software engineering tasks that require deep understanding of code structure, programming language semantics, and software design principles.

**Problem Statement**

```
`PolyElement.as_expr()` not accepting symbols
The method `PolyElement.as_expr()`

https://github.com/sympy/sympy/blob/193e3825645d93c73e31cdceb6d742cc6919624d/sympy/polys/rings.py#L618-L624

is supposed to let you set the symbols you want to use, but, as it stands, either you pass the wrong number
of symbols, and get an error message, or you pass the right number of symbols, and it ignores them, using
`self.ring.symbols` instead:

```python
>>> from sympy import ring, ZZ, symbols
>>> R, x, y, z = ring("x,y,z", ZZ)
>>> f = 3*x**2*y - x*y*z + 7*z**3 + 1
>>> U, V, W = symbols("u,v,w")
>>> f.as_expr(U, V, W)
3*x**2*y - x*y*z + 7*z**3 + 1
```
```

Figure 15: Problem statement presented to the agent, describing the issue that needs to be resolved.

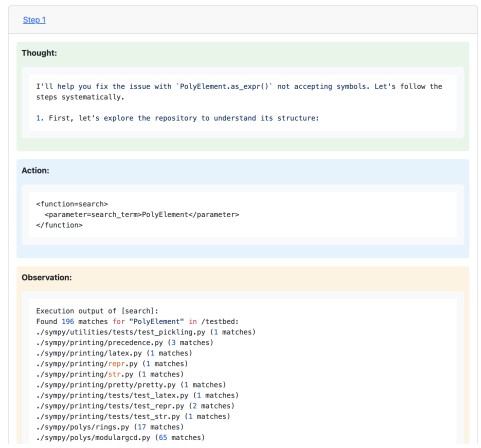

(a) Step 1: Initial analysis and exploration

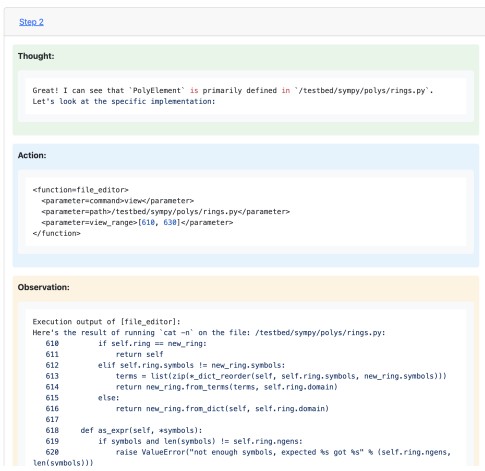

(b) Step 2: Detailed Exploration

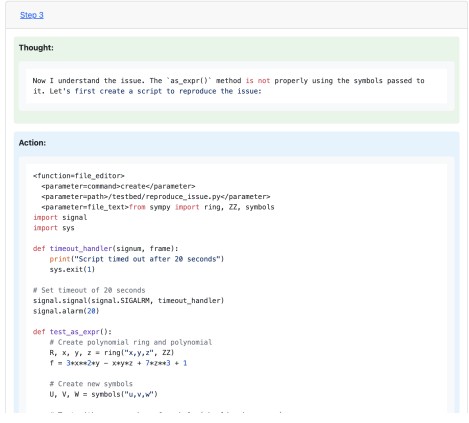

(c) Step 3: Reproducing the issue

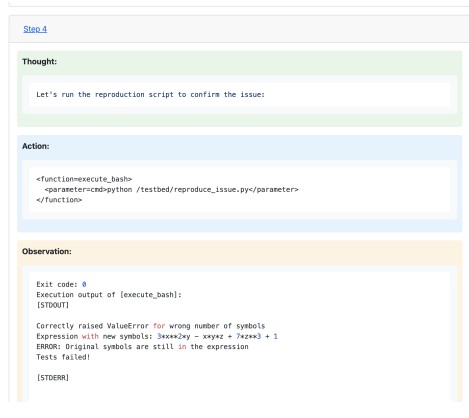

(d) Step 4: Running reproduction tests

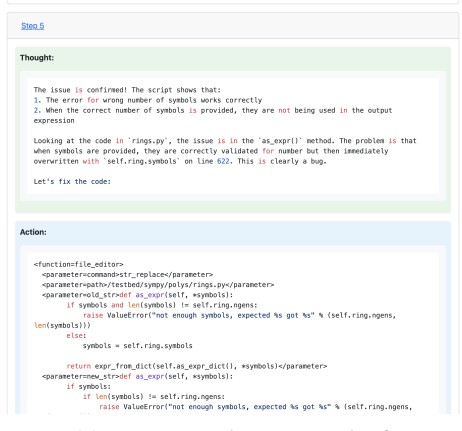

(e) Step 5: Implementing the fix

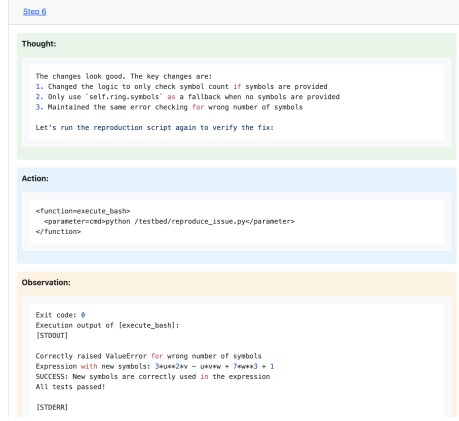

(f) Step 6: Verifying the fix

Figure 16: Short successful agent trajectory (using our 32B model) showing the step-by-step process of solving a software engineering task. The agent analyzes the problem, identifies the root cause, implements a solution, tests it, and verifies that it resolves the issue.

