# OpenReview forum: "R2E-Gym: Procedural Environment Generation and Hybrid Verifiers for Scaling Open-Weights SWE Agents"
_colmweb.org/COLM/2025/Conference — COLM 2025_

### Official Review · Reviewer_gmVU · 2025-05-13

**Rating:** 7
**Confidence:** 4
**Ethics Flag:** 1

**Summary:**

This paper presents AgentGym, a resource that is the largest of its kind and breaks new ground in terms of proposing two complementary metrics (execution based and execution free) that when combined are argued to achieve comparable performance on SWE-Bench with proprietary models.  Many of the technical details are in the appendix.  A promise is made that the code and resources will be opensourced when the paper is accepted.

**Questions To Authors:**

I'd like to hear an explanation for how to reconcile these two things.  I'd also like to see it acknowledged that though the results of this paper seem to indicate value in integrating execution-based and execution-free metrics, that past studies cast doubt of the validity of execution-free metrics.  Here's a reference to corroborate that:

Xie et al. CodeBenchGen: Creating Scalable Execution-based Code Generation Benchmarks (on arxiv)

Can you shed some light on how it's possible that integrating a low-validity metric improves performance?  What kinds of cases does it help with over-and-above just the execution-based metric?

**Reasons To Accept:**

The paper includes a large amount of experimentation to locate where the techniques fall in the landscape.  The experimentation is rigorous, and SOTA models are used.  The claims to fame are the size of the resource and the performance of the hybrid approach being in league with that of proprietary models.  As I was reading, I felt that the paper was vague -- for example, when backtranslation is first mentioned, what isn't mentioned is what is being translated to what.  So the assertion is not really understandable until a little later when the idea of backtranslation for this application is unpacked.  The appendix make a lot more clear, so then I was less concerned about this.

**Reasons To Reject:**

I am not totally convinced of the claim that the opensource approach is comparable to the best proprietary models.  Figure 1c seems to make that case, but Table 3 (which is never referenced in the paper) paints a different picture.  Here the best result from the opensource approach is comparable to middling proprietary models, but not the best ones.

Apart from the good performance numbers, the large number of experiments, and the bigness of the resources provided, the work is not that creative, and it's not clear that it really pushes forward the science.  I'd like to hear more about what we understand better from this work and not just what makes the performance numbers go up.

---

> ### Author Response · Authors · 2025-06-03
>
> We thank the reviewer for their positive feedback and are pleased that they found our breaking new ground for test-time scaling and with rigorous experimentation, including those with SOTA models.
>
> > I am not totally convinced of the claim that the open-source approach is comparable to the best proprietary models. Figure 1c seems to make that case, but Table 3 (which is never referenced in the paper) paints a different picture. Here the best result from the opensource approach is comparable to middling proprietary models, but not the best ones.
>
> Thanks. Figure 1c compares our system, comprising a 32B model, against strong closed-source model offerings from OpenAI (o1 models) and Anthropic (Claude-3.6-Sonnet). Notably, Claude-3.6-Sonnet was SOTA on the SWE-Bench leaderboard for 4 months starting November until February-end (1 month before COLM submission) when Claude-3.7-Sonnet was released.
>
> We only claim comparable performance to the Claude-3.6-based systems and provide a comparison to Claude-3.7-based systems in Table 3 for completeness. We will modify the draft to make this clearer. We would, however, like to point out that Claude-3.6 and OpenAI o1 models are notably strong closed model systems, and our work makes progress towards achieving parity with them.
>
>
>
> $\newline$
>
>
> > I'd like to hear an explanation for how to reconcile these two things. I'd also like to see it acknowledged that though the results of this paper seem to indicate value in integrating execution-based and execution-free metrics, that past studies cast doubt of the validity of execution-free metrics. Here's a reference to corroborate that:
> >
> > Xie et al. CodeBenchGen: Creating Scalable Execution-based Code Generation Benchmarks (on arxiv)
>
> Thanks for the question.
>
> **Indeed, in Sec. 4.2, we also observe limitations of execution-free verifiers.** We perform various analyses to understand the reason for their limitations, and one of our key findings is that execution-free verifiers can be unreliable as they may be biased by other heuristics (e.g., agent thoughts) over the final output patch. So, for example, as depicted in Fig. 8, even for an incorrect trajectory, the verifiers can get biased by agent thoughts (showing confidence) and predict SUCCESS for an incorrect trajectory. But even though execution-free verifiers are imperfect, they can still be useful and provide some signal for comparing correct and incorrect programs, as also found in various previous works such as SWE-Gym [5], Neural Code Rerankers [6].
>
> The reason that execution-free verifiers provide value in combination with execution-based verifiers is that, unlike CodeBenchGen [4], **our execution-based verifier (or testing agent) uses self-generated tests and does not have access to ground truth information including the sourcecode or tests. Thus our generated tests can differ from oracle tests that are found in the repository and can be imperfect**. We again study the limitations of these tests and highlight two key limitations – lack of distinguishability (i.e., lacking coverage to distinguish correct patches from incorrect patches) and generating toxic tests (i.e., tests that pass incorrect patches but fail correct patches). Through this analysis (Sec. 4), we demonstrate that combining the two weak verifiers can result in improved performance of the overall system.
>
> * For example, distinguishability - in practice, the agent-generated tests do not cover all edge or challenging test cases; thereby leading to ties during verification. The use of an execution-free verifier can help provide better distinguishability in such cases; given the agent trajectory and output patch, the execution-free verifier is able to easily verify agent trajectories where the agent tried to handle all edge cases beyond just the problem statement.
>
>
>
>
>
> $\newline$
>
> > Can you shed some light on how it's possible that integrating a low-validity metric (execution-free) improves performance? What kinds of cases does it help with over-and-above just the execution-based metric?
>
>
> Great question! As noted above, **we find that integrating execution-free and execution-based verifiers is useful when execution-free verifier scores provide distinguishability in case of a tie with execution-based test scores.**
>
> To elaborate:
>
> * Ideally, given a very good test-generation agent, execution-based verification should alone be sufficient to rerank the generated patches.
> * However, in practice, the testing agent-generated tests suffer from low distinguishability and do not cover all edge or challenging test cases; thereby leading to ties during verification.
> * The use of execution-free verifiers can help provide better distinguishability in such cases; as given the agent trajectory and output patch, the execution-free verifier is able to easily verify agent trajectories where the agent tried to handle all edge cases beyond just the problem statement.

---

> > ### Author Response · Authors · 2025-06-03
> >
> > > How it really pushes forward the science ... I'd like to hear more about what we understand better from this work ...
> >
> >
> >
> > As also noted by [fPRC,2kAH, DvQH], the paper proposes two key insights for pushing forward scaling of open-weight SWE agents:
> >
> > 1. **Synthetic data (instead of human PRs) can enable more scalable SWE-agent training**:
> >    * Advancing the development of open-weight SWE agents (using SFT/RFT/RL) faces a fundamental challenge from the data perspective: *scalable curation of high-quality execution environments to train these models*.
> >    * While prior works collect executable test environments, their sample size and diversity of curated data are limited by their reliance on human-written issues and test cases. **In this paper, for the first time we show that synthetic data curation can enable collecting large amounts of high-quality executable environments, enabling more scalable agent training.**
> >    * Please note that scalable synthetic data generation approaches have been very successful and used by various top labs. Even within the coding domain, Magicoder [2] (a synthetic data curation approach) has been adopted for scalable data curation for training some of the SOTA models, including Google Gemma [1], Meta LLama [3].  In this work, for the first time, we show that synthetic data curation can also be applied for the SWE Agent domain, enabling more scalable agent training.
> >
> > 2. **Hybrid test-time scaling: - Execution-based and Execution-free verifiers have complementary strengths and weaknesses which can be leveraged to get better performance**.
> >    * Prior works often study execution-based and execution-free verifiers in isolation and lack in-depth analysis on their working mechanisms.
> >    * In this paper, we provide a unique and in-depth analysis, showing that these two approaches actually show complementary strengths (refer L50-64, Sec. 4.2, 4.3) which can be leveraged for getting significantly larger gains as opposed to each of them separately.
> >    * In particular, we show that while execution-free verifiers are imperfect and can be biased by heuristics, execution-based verifiers can suffer from low distinguishability between generated solutions and toxic test generation. As discussed above, while execution-free verifiers by themselves are imperfect, they can be used alongside execution-based verifiers for providing distinguishability in case of a tie with execution-based test scores (refer Sec. 4.3); thereby leading to much better performance when scaling test-time compute.
> >
> >
> > ---
> >
> > **References**:
> >
> > [1] Gemma: Open Models Based on Gemini Research and Technology, Gemma Team, 2024
> >
> > [2] Magicoder: Empowering Code Generation with OSS-Instruct, Wei et al., ICML 2024
> >
> > [3] The Llama 3 Herd of Models, Meta AI, 2024.
> >
> > [4] CodeBenchGen: Creating Scalable Execution-based Code Generation Benchmarks, Xie et al., 2024
> >
> > [5] Training Software Engineering Agents and Verifiers with SWE-Gym, Pan et al., ICML 2025
> >
> > [6] Fault-Aware Neural Code Rankers, Inala et al., NeurIPS 2022

---

> > > ### Comment · Reviewer_gmVU · 2025-06-07
> > > **thanks for the clarifications/elaboration**
> > >
> > > Thanks, this sounds good.  I'll keep my original score for now.

---

> > > > ### Author Response · Authors · 2025-06-09
> > > >
> > > > Thanks for taking the time to read our response. We are happy that our response addressed your concerns. Please let us know if there are any further questions or concerns that we can help address to improve your score.

---

### Official Review · Reviewer_DvQh · 2025-05-14

**Rating:** 8
**Confidence:** 4
**Ethics Flag:** 1

**Summary:**

This work introduces AgentGym, a large set of procedurally generated executable environments (8.7K tasks) for training SWE agents. This is done using test and instruction generation from the repository commits directly, skipping the reliance on human-written GitHub Issues and PRs.

In particular, the first GitHub Python repositories that can be built successfully. They then filter the commits with some heuristics. And just based on that, they produce an entire SWE task+evaluation with an executable environment. The tests are often LLM-generated, and the task instruction is generated from the commit changes.

They show that training a model on this data leads to substantial improvements in SWEBENCH-VERIFIED.

Additionally, in this environment, they explore two test-time scaling strategies: execution-based (passing tests) and execution-free (styling, process-based), and find their complementary strengths that can be combined to obtain even better results.

This work also achieves SOTA of 51% on the SWE-Bench Verified using open models.

**Reasons To Accept:**

One of the main bottlenecks for training Swe agents is curating large-scale executable environments. This work introduces a pipeline and approach to scaleably generate these, and also demonstrates its success by developing a SOTA swe agent.

The experiments and contribution for demonstrating the complementarity of the two test-time scaling approaches are also a novel and interesting finding in themselves.

**Reasons To Reject:**

This is an excellent and much-needed work! I don't see any strong reason to reject this paper. A couple of minor points:

Given LLMs are used to generate the data. It'd be good to have a sense of the quality and solvability of these tasks. E.g., to what extent can humans solve these auto-generated tasks? I understand, though, this is for training, so the bar is not necessarily as high, and downstream performance improvement already is a good signal of utility.

It is limited to only Python. But the entire pipeline can, in the future, perhaps be extended to other languages.

---

> ### Author Response · Authors · 2025-06-03
>
> We thank the reviewer for their positive feedback and are excited that they found our work excellent and much-needed with novel and interesting findings.
>
> > This is an excellent and much-needed work! I don't see any strong reason to reject this paper. A couple of minor points:
> >
> > Given LLMs are used to generate the data. It'd be good to have a sense of the quality and solvability of these tasks. E.g., to what extent can humans solve these auto-generated tasks? I understand, though, this is for training, so the bar is not necessarily as high, and downstream performance improvement already is a good signal of utility.
>
> Please note that while LLMs are used to generate data, the problems are generated directly from commits (instead of human PRs). Thus, each problem should be solvable given two conditions:
>
> 1) The ground truth patch (commit patch) is valid, i.e., some tests fail before application of the ground truth patch, and all tests pass after application of the ground truth patch.
> 2) The generated natural language problem statement is non-ambiguous / solvable: i.e., the repository and the natural language problem statement provide complete information required for fixing the issue.
>
> Regarding (1), we adopt our test validation strategy (discussed in Sec. 2 and Appendix A), ensuring that each ground-truth patch is validated in the training dataset. Regarding (2), to ensure non-ambiguity in the natural language problem statement, we use the backtranslation approach (refer Sec. 2) for generating Github issues grounded in the execution behavior (execution outputs) of the program.
>
> Further evaluation: **While exhaustive human annotations were not feasible due to large sample size and resource constraints, we also performed LLM as Judge annotations to evaluate the quality of the curated problems. In particular, we follow the annotation criteria from [1], where for each problem, we collect annotations on the two axes for problem quality.** We use a total of 500 problems for both synthetic and real Github tasks.  o4-mini was used for LLM as judge experiments (note that o3-mini was used as the back-translation model).
>
> * **Are the tasks well specified?**: Since SWE-agents are required to generate an output patch given a problem statement and the codebase, we wish to have non-ambiguous problem statements to ensure the solvability of the task. To this end, following [1], we label the problem statement with these 4 possible labels:
>   - 0: The issue is well-specified and it is clear what is required for a successful solution.
>   - 1: There are some blanks to fill in about the issue, but there is a sensible interpretation of what is required for a successful solution.
>   - 2: The issue is vague and there is room for ambiguity. It is unclear what a successful solution would look like.
>   - 3: It is almost impossible to understand what you are being asked to do without further information.
> * **How valid are the evaluation criteria (i.e. tests)?** To further evaluate the quality of the generated tests, we label the tests with these 4 possible labels:
>   - 0: The tests perfectly cover all possible solutions.
>   - 1: The tests cover the majority of correct solutions, however some unusual solutions may be missed.
>   - 2: The tests work but some perfectly reasonable solutions may be missed by the tests.
>   - 3: The tests are too narrow/broad or they look for something different than what the issue is about.
>
> Results are shown below. As seen, we observe that synthetic problems not only perform similarly w.r.t. real github issues after downstream training (Sec. 3.1) but can also help control / reduce the ambiguity in problem statements when creating new SWE tasks.
>
>
>
> | Performance                                                  | Synthetic Problems | ReaL Github Issues |
> | ------------------------------------------------------------ | :----------------: | :----------------: |
> | Downstream Performance [Sec. 3.1] ($\uparrow$)                 |       27.8%        |       28.0%        |
> | Are tasks well specified? ($\downarrow$)                       |  0.122 $\pm$ 0.362   |  0.632 $\pm$ 0.893   |
> | How valid are the evaluation criteria (i.e. tests)? ($\downarrow$) |  1.024 $\pm$ 1.205   |         -          |
>
>
>
>
> We will include this evaluation in the updated version of our paper
>
>
>
>
>
>
>
>
> $\newline$
>
>
> > It is limited to only Python. But the entire pipeline can, in the future, perhaps be extended to other languages.
>
> Thanks! While the current work mostly focuses on synthetic data curation for Python repositories, the proposed testing and back-translation framework for synthetic SWE-task generation is generalizable to other languages.
>
> **References**:
>
> [1] Introducing SWE-bench Verified, Chowdhury et al., OpenAI, 2024

---

> ### Author Response · Authors · 2025-06-09
>
> As the end of the response period is approaching, we would like to check in with the reviewer to ensure that our clarifications adequately address your questions. Please let us know if you need more information for making your final decisions!

---

### Official Review · Reviewer_2kAH · 2025-05-20

**Rating:** 9
**Confidence:** 4
**Ethics Flag:** 1

**Summary:**

This paper proposes AgentGym, consisting of the largest yet synthetic dataset SynGen of annotated software engineering (SWE) tasks of the SWE-Bench type, and a hybrid verification method that leads to SOTA performance on SWEBench-Verified. The paper is solid --- very well written, with the right level of details in the main paper and a great choice of experiments and articulation of findings. This should serve as a valuable resource for work on building open agents. Further, the insight that execution-based vs. execution-free verifiers offer complementary benefits is super useful in this line of work.

**Questions To Authors:**

N/A

**Reasons To Accept:**

* Well-written paper, was enjoyable to read.

* Very timely and useful resource for several groups trying to build strong agents.

* Valuable insight that execution-based and execution-free verifiers offer complementary benefits and that this can be leveraged to create a more powerful agent.

* Strong performance of the resulting trained model on SWEBench-Verified.

**Reasons To Reject:**

(**Important but not a reason to reject**) Please note ``AgentGym'' already exists as a related but apparently different resource for agents. See https://agentgym.github.io/ and https://arxiv.org/abs/2406.04151

---

> ### Author Response · Authors · 2025-06-03
>
> We thank the reviewer for their positive feedback and are excited that they found are work well-written and enjoyable to read, very timely and super useful, including a great choice of experiments and articulation of findings with valuable insights and strong performance.
>
> > (**Important but not a reason to reject**) Please note ``AgentGym'' already exists as a related but apparently different resource for agents. See https://agentgym.github.io/ and https://arxiv.org/abs/2406.04151
>
> Thanks! We will update the paper title to avoid any naming conflict.

---

> > ### Comment · Reviewer_2kAH · 2025-06-09
> > **Re: Official Comment by Authors**
> >
> > Other than avoiding the naming conflict, I don't have any further suggestions.
> >
> > Congratulations on a well-written and nicely executed piece of work!

---

> ### Author Response · Authors · 2025-06-09
>
> As the end of the response period is approaching, we would like to check in with the reviewer to ensure that our clarifications adequately address your questions. Please let us know if you need more information for making your final decisions!

---

### Official Review · Reviewer_fpRC · 2025-05-21

**Rating:** 8
**Confidence:** 2
**Ethics Flag:** 1

**Summary:**

The paper introduces AgentGym, the largest procedurally curated environment, consisting of 8.7k+ problems (>3x increase over prior benchmarks), with executable gym environments, unit tests, and task descriptions, for training SWE-agents. The paper proposes SynGen: a synthetic data curation recipe for curating executable training environments w/o relying on human tests and issues -- execution environments can directly be curated from commits through backtranslation. The paper also investigates "Hybrid inference time scaling": 1) execution-based verifiers that evaluate patches through test cases. 2) execution-free verifiers that assess trajectories through learned models. Empirically shows that a hybrid approach performs better than either alone. The paper is well written and easy to follow.

Overall, the paper makes a significant contribution by scaling up the synthetic SWE task generation and showing strong empirical results of different verifier strategies. I recommend an accept.

**Reasons To Accept:**

* Empiricism, Data, and Evaluation

The paper provides extensive compelling empirical results to support their claims. The paper shows competitive open-weight performance (BEST@26 w/ hybrid 51% on SWEBench-Verified), closing the gap with proprietary systems (Claude-3.7 Sonnet at 70.3% BEST@Any).

* Technological Impact

The paper claims to open source their environments (more than 8.7k tasks), models, and agent trajectories.

* Understanding Depth, Principled Approach

The paper provides insights in the complementary merits of execution-based verifiers vs execution-free verifiers. The paper provides thorough analysis on test-time scaling approaches.

* Clarity, Honesty, and Trust

The paper is well written and mostly easy to follow.

**Reasons To Reject:**

* Understanding Depth, Principled Approach

I'd be curious to learn how to evaluate the quality of the curated problems--a qualitative evaluation of the curated benchmark would be helpful.

The draft offers few insights into common failure modes of the proposed approach (e.g. types of bugs still missed, qualitative examples).

The reasoning behind choosing Top­n in the hybrid verifier (Eqn. 2) and its sensitivity is not deeply examined.

* Clarity, Honesty, and Trust

Consider describe the backtranslation approach in the paper.

Some acronyms are introduced before the definition (4.1, eq. 1).

---

> ### Author Response · Authors · 2025-06-03
>
> We thank the reviewer for their positive feedback and are pleased that they found our work well-written, a significant contribution, and supported by extensive, compelling empirical results.
>
> > I'd be curious to learn how to evaluate the quality of the curated problems--a qualitative evaluation of the curated benchmark would be helpful.
>
> ### Quantitative Results
>
> * **Downstream Evaluation.** In Section 3.1 (L143-150) we study the quality of the curated problems by evaluating their downstream effect on training. Specifically, we compare models trained on real GitHub issues with models trained on our synthetic problems, using the same number of trajectories (400) for each. Models trained on synthetic data achieve nearly identical performance (27.8 % vs. 28.0 %), demonstrating that our procedurally generated environments match the training value of real-world examples.
>
> * **LLM-as-Judge Annotations.** While exhaustive human annotations were not feasible due large sample size and resource constraints, we performed a LLM-as-Judge protocol to asses the problem quality. In particular, we follow the annotation criteria from [1], where for each problem, we collect annotations on the following axis for problem quality. Following [1], we annotate 500 synthetic and 500 real GitHub tasks along two axes:
>
>   * **Are the tasks well specified?** Since SWE-agents require to generate output patch given problem statement and the codebase, we wish to have non-ambigous problem statements to ensure solvability of the task. To this end, following [1], we label the problem statement with 4 possible labels:
>
>     * 0: The issue is well-specified and it is clear what is required for a successful solution.
>
>     - 1: There are some blanks to fill in about the issue, but there is a sensible interpretation of what is required for a successful solution.
>
>     - 2: The issue is vague, and there is room for ambiguity. It is unclear what a successful solution would look like.
>
>     - 3: It is almost impossible to understand what you are being asked to do without further information.
>
>   *  **How valid are the evaluation criteria (i.e., tests)?** To further evaluate the quality of the generated tests, we label the tests with these 4 possible labels:
>       * 0: The tests perfectly cover all possible solutions.
>       * 1: The tests cover the majority of correct solutions, however, some unusual solutions may be missed.
>       * 2: The tests work but some perfectly reasonable solutions may be missed by the tests.
>       * 3: The tests are too narrow/broad or they look for something different than what the issue is about.
>
>
>
>
> Results are shown below. As seen, we observe that synthetic problems not only perform similarly w.r.t. real github issues after downstream training (Sec. 3.1) but can also help control / reduce the ambiguity in problem statements when creating new SWE tasks.
>
> | Performance                         | Synthetic Problems | ReaL Github Issues |
> | ------------------------------------------------------------ | :----------------: | :----------------: |
> | Downstream Performance [Sec. 3.1] [$\uparrow$]       |    27.8%    |    28.0%    |
> | Are tasks well specified? [$\downarrow$]         | 0.122 $\pm$ 0.362 | 0.632 $\pm$ 0.893 |
> | How valid are the evaluation criteria (i.e. tests)? [$\downarrow$] | 1.024 $\pm$ 1.205 |     -     |

---

> > ### Author Response · Authors · 2025-06-03
> >
> > (continued)
> > ### Qualitative Results
> >
> > In addition to the above quantitative evaluation, we also provide some qualitative examples, comparing the quality of human issues vs synthetically generated problem statements. While human-written issues can have imperfections arising from ambiguity or a lack of clarity, we find our execution assisted back-translation approach leads to high-quality and descriptive issues.
> >
> > SWE-Bench `scikit-learn__scikit-learn-14520` issue (analysis borrowed from [1])
> >
> > ```
> >
> > Copy param ignored in TfidfVectorizer
> >
> > I was playing with vectorizers and I found this:
> >
> > https://github.com/scikit-learn/scikit-learn/blob/ae16319626e2ca6ca0e54d4a5b83f73f817232aa/sklearn/feature_extraction/text.py#L1669
> >
> > However that parameter is not used later in the method.
> >
> > Here `copy=False` is used:
> >
> > https://github.com/scikit-learn/scikit-learn/blob/ae16319626e2ca6ca0e54d4a5b83f73f817232aa/sklearn/feature_extraction/text.py#L1692
> >
> > Is there anything I am missing?
> >
> > ```
> >
> >
> > Here, the issue says that the `copy` argument is not used; however does not specify what the expected behavior should be, i.e., should the copy argument be used or removed. **This ambiguity leads to even “correct” solutions getting penalized.**
> >
> > $\newline$
> >
> > **Our backtranslated issues, on the other hand, follow a good issue structure where they use code snippets to describe the bug along with a bug report (observed behavior) and explain what the expected behavior should be**. Please see the following example (from the appendix)
> >
> > ```md
> >
> > ** Title :** Calling ` load () ` Before ` draft () ` Causes ` draft () ` to Fail for
> >
> > JPEG Images
> >
> > ** Description :**
> >
> > When generating a thumbnail for a JPEG image using the ` thumbnail () ` method , the method calls ` load () ` before ` draft () `. This sequence results in the ` draft () ` method returning `None `, which prevents the thumbnail from being properly optimized .
> >
> > ** Example Code :**
> >
> > \``` python
> >
> > from PIL import Image
> >
> > with Image . open (" Tests / images / hopper . jpg ") as im :
> >
> >     im.thumbnail ((64 , 64) )
> >
> > \```
> >
> > ** Expected Behavior:**
> >
> > The `thumbnail()` method should utilize the `draft()` method to optimize the image size before loading, ensuring that the thumbnail is resized correctly and efficiently.
> >
> > ** Actual Behavior:**
> >
> > The `draft()` method returns `None` because `load()` is invoked before it. This prevents the thumbnail from being optimized, potentially leading to incorrect thumbnail sizes or unnecessary memory usage
> >
> > ```
> >
> >
> >
> > $\newline$
> >
> > > The draft offers few insights into common failure modes of the proposed approach (e.g. types of bugs still missed, qualitative examples).
> >
> > As discussed in Sec. 4, we describe limitations of both execution-based verifiers and execution-free verifiers. In particular, execution-based verifiers struggle with a lack of distinguishability (i.e., lacking coverage to distinguish correct patches from incorrect patches) and generating toxic tests (i.e., tests that pass incorrect patches but fail correct patches). Execution-free verifiers, on the other hand, have strong distinguishability but rely on possible heuristic approaches to re-rank the generations.
> >
> > While the two verifiers complement each other's weaknesses, they do not mitigate all of them. For example, in cases where execution-based verifiers do not provide distinguishability, we rely on execution-free verifiers, which may still be incorrect and do not provide any improvements. Similarly, execution based verifiers can still struggle with generating toxic tests, which will select incorrect solutions irrespective of execution-free verifiers.
> >
> > **We additionally study where our test generation fails and provide some qualitative examples (for brevity, three examples) below:**
> >
> > 1. For [Django issue](https://code.djangoproject.com/ticket/30776), the fix involves setting the `maxLength` attribute to [254](https://github.com/django/django/commit/6c9778a58e4f680db180d4cc9dc5639d2ec1b40c). The model implementation instead sets it to “254” (string 254) but also writes a corresponding test to check for “254”, thus producing a toxic test that fails to rank the correct solution.
> > 2. Django tests use sqlite database for writing tests. SQLite has a unique quirk where 3.14 % 10 is rounded off to 3. Our trained models do not know this and instead generate both solution and patch to return 3.14, leading to a toxic test.
> > 3. Sympy performs imperfect and approximate algebraic manipulations. For one SWE-Bench problem, the solution does not solve all cases of the problem, but only some of the common ones. The model generated tests instead expect the more general solution and fail to distinguish the correct and incorrect solutions. [Issue 15017](https://github.com/sympy/sympy/pull/15017) faces similar concerns, as evident by a long chain of discussion in the issue thread. Thus, ambiguity can lead to imprecise verification.
> >
> > Thanks for raising this. We will add this discussion to the final version.

---

> > ### Author Response · Authors · 2025-06-03
> >
> > > The reasoning behind choosing Top­n in the hybrid verifier (Eqn. 2)
> >
> > We study the role of Top-n and sensitivity in the hybrid verifier (Eq. 2) through quantitative ablations and analysis in Sec. 4.4 (L282) and Figure 6-right. Reasoning: **We find that use the top-n filtering helps mitigate the impact of toxic tests (Sec. 4.2) by restricting the application of execution-based verification to higher-quality patches (identified via execution-free reward scores).**
> >
> > To elaborate, we find that using the top-n filtering helps mitigate the impact of toxic tests (i.e., tests that pass incorrect patches but fail correct patches described in Sec. 4.2). Particularly, since the top-n aggressively filters the lowest ranked patches from the execution-free verifiers, it can help by removing the incorrect patches that “correlate” with the incorrect tests thus getting re-ranked the highest.
> >
> >
> > $\newline$
> >
> > > Consider describe the backtranslation approach in the paper.
> >
> > Thanks. **We will amend the paper, adding more details on the back-translation approach.** For reference, please also find the details below:
> >
> > Prior works [2] rely on back-translating GitHub programs to generate natural language problem descriptions corresponding to the programs. However, naively back-translating code changes is quite noisy as models often generate generic problem statements that do not capture the essence of the code changes. Instead, we identify that human-written issues often contain failing tests and execution traces as part of bug reports. We use this observation and use the test execution behavior (test programs, failing tests, and their execution outputs) in the back-translation process and ask the language models to provide a concise description of the failing test when forming the GitHub issue. Similar to existing works [3,4], we find that using test execution information allows generating precise and directed problem statements. Please refer to Appendix A for exact prompts used.
> >
> > **References**:
> >
> > [1] Introducing SWE-bench Verified, Chowdhury et al., OpenAI, 2024
> >
> > [2] Magicoder: Empowering Code Generation with OSS-Instruct, Wei et al., ICML 2024
> >
> > [3] R2e: Turning any github repository into a programming agent environment, Jain et al., ICLR 2024
> >
> > [4] BigCodeBench: Benchmarking Code Generation with Diverse Function Calls and Complex Instructions, Zhuo et al., ICLR 2025

---

> ### Author Response · Authors · 2025-06-09
>
> As the end of the response period is approaching, we would like to check in with the reviewer to ensure that our clarifications adequately address your questions. Please let us know if you need more information for making your final decisions!

---

> > ### Comment · Reviewer_fpRC · 2025-06-10
> > **Thanks for the detailed responses**
> >
> > I thank the authors for the detailed responses. I will keep my rating of "8: Top 50% of accepted papers, clear accept".

---

### Author Response · Authors · 2025-06-03
**General Response**

We sincerely thank the reviewers for their positive feedback and are delighted that **all reviewers** regard our work as a significant contribution, supported by extensive experiments and strong performance in scaling open-weight SWE agents. Furthermore, reviewers found the paper well-written and enjoyable to read [fPRC, 2kAH], offering valuable insights [fPRC, 2kAH, DvQH], representing excellent and much-needed work [DvQH, 2kAH], and presenting rigorous experiments [fPRC, 2kAH, DvQH, gmVU].

We hope the additional results and clarifications below address all questions raised during the discussion period.

---

### Decision · Program_Chairs · 2025-07-08

**Decision:**

Accept

**Comment:**

The paper presents two ideas for improving Software Engineering (SWE) agents: 1) a scalable method for synthetic curation of execution environments, and 2) a way for combining execution-based and execution-free verifiers to improve agent performance. The experimental results clearly demonstrate the utility of the methods in improving performance on the SWE-bench tasks. The paper is well written overall and the experiments are well executed.

The reviewers have provided areas for improvement including details on certain aspects of the methodology, additional assessment of the quality of the curated dataset, and some error analysis. The authors are encouraged to address these in the camera ready. Their response does a good job of this.